# What Drives Paper Acceptance? A Process-Centric Analysis of Modern Peer Review

## Abstract

Peer review is the primary mechanism for evaluating scientific contributions, yet prior studies have mostly examined paper features or external metadata in isolation. The emergence of open platforms such as OpenReview has transformed peer review into a transparent and interactive process, recording not only scores and comments but also rebuttals, reviewer–author exchanges, reviewer disagreements, and meta-reviewer decisions. This provides unprecedented *process-level* data for understanding how modern peer review operates. In this paper, we present a large-scale empirical study of ICLR 2017–2025, encompassing over 28,000 submissions. Our analysis integrates four complementary dimensions, including the structure and language quality of papers (e.g., section patterns, figure/table ratios, clarity), submission strategies and external metadata (e.g., timing, arXiv posting, author count), the dynamics of author–reviewer interactions (e.g., rebuttal frequency, responsiveness), and the patterns of reviewer disagreement and meta-review mediation (e.g., score variance, confidence weighting). Our results show that factors beyond scientific novelty significantly shape acceptance outcomes. In particular, the rebuttal stage emerges as a decisive phase: timely, substantive, and interactive author–reviewer communication strongly increases the likelihood of acceptance, often outweighing initial reviewer skepticism. Alongside this, clearer writing, balanced visual presentation, earlier submission, and effective resolution of reviewer disagreement also correlate with higher acceptance probabilities. Based on these findings, we propose data-driven guidelines for authors, reviewers, and meta-reviewers to enhance *transparency* and *fairness* in peer review. Our study demonstrates that process-centric signals are essential for understanding and improving modern peer review.

## 1 Introduction

The introduction of OpenReview has transformed peer review at major AI/ML conferences from a closed, one-way evaluation into a transparent and interactive process (Tran et al., 2020). With the public release of reviews, reviewer confidence levels, author rebuttals, reviewer discussions, and meta-review decision statements, researchers now have access to ***process-level data*** that were previously unavailable. This development enables systematic investigation of peer review dynamics beyond outcome-based measures (Ragone et al., 2011). Building on this modern peer review system, this study seeks to explain the determinants of final acceptance (accept/reject) from a process-centric rather than an outcome-centric perspective (Shah, 2022).

Prior research has largely relied on *outcome* indicators such as acceptance decisions or citation counts, and often focused on a single year or a limited set of features (Garcia-Costa et al., 2022). Moreover, there has been no integrated framework that simultaneously considers interaction logs (e.g., rebuttals and reviewer discussions), external signals (e.g., arXiv posting, code and data release, submission timing (Xie et al., 2024)), and stylistic factors (e.g., writing style, visual elements (Crossley & Roscoe, 2014; Kuznetsov et al., 2024)). Even though the spread of OpenReview has made these data systematically available (Zahorodnii et al., 2025; Idahl & Ahmadi, 2025; Wang et al., 2021), comprehensive studies that connect them to explain the dynamics of peer review remain limited. To address this gap, our work integrates process signals, external signals, and stylistic factors into a unified analytical framework.

Our analysis draws on the complete OpenReview record of ICLR 2017–2025, combining reviews, meta-reviews, rebuttal and interaction threads, reviewer scores and confidence, submission and deadline timestamps, arXiv postings, and code/data disclosures. We also parsed manuscript PDFs to extract document structure and linguistic features, including table/figure/equation density, readability and lexical diversity, and reference recency. The corpus spans 28,358 submissions, growing from 490 in 2017 to 8,629 in 2025–an approximately 17-fold increase–and uniquely integrates process logs, external signals, and stylistic elements for longitudinal large-scale analysis.

This study addresses four central research questions: 1) Reviewer Disagreement & Mediation: How do score variance, reviewer confidence, and sentiment affect acceptance outcomes? 2)Review Interaction Logs: How do the speed, length, depth, and frequency of rebuttal exchanges influence acceptance? 3)Submission Strategies & External Metadata: How do submission timing, arXiv posting, and the release of code/data function as acceptance signals? 4)Visual, Linguistic, and Reference Patterns: How do tables, figures, equations, readability, and reference recency correlate with contribution and novelty evaluations?

Our contributions are threefold. First, we present a *process-centric framework* linking reviewer disagreement, interactions, external signals, and stylistic patterns to explain acceptance decisions. Second, we construct and release a longitudinal dataset (ICLR 2017–2025) integrating OpenReview logs, arXiv metadata, and PDF-derived features. Third, we provide *practical guidelines* with recommendations for authors, meta-reviewers, and organizers to ensure fairness and transparency.

The remainder of the paper is organized as follows. Section 2 defines the modern peer review system and describes our data and metrics. Sections 3–6 analyze reviewer disagreement, interaction logs, submission strategies and external metadata, and visual/linguistic/reference patterns, respectively. The final discussion integrates findings, explores limitations, and outlines future research directions. A detailed survey of related work is provided in Appendix A.

## 2 MODERN PEER REVIEW: DEFINITION, RESEARCH QUESTIONS, AND DATASET

Peer review has long been central to evaluating scientific work, but traditional systems relied on closed evaluation and **one-way** communication. Since 2013, with the adoption of platforms like OpenReview, AI/ML conferences such as ICLR have pioneered transparent and interactive peer review. Reviews, rebuttals, reviewer discussions, and meta-reviews are now publicly accessible, transforming the process into structured, analyzable data. This chapter defines the scope of modern peer review, outlines our research questions, and introduces the dataset used in this study.

**Defining the f Peer Review System.** We define the modern system as one characterized by **transparency** (open visibility of reviews and decisions) and **interactivity** (author rebuttals, reviewer debates, meta-review mediation). These features convert the review process itself–once invisible–into *process-level* data that can be systematically analyzed.

**Key Dimensions and Research Questions.** Building on ICLR 2017–2025 data, we formulate four sets of research questions:

1. **Reviewer Disagreement & Mediation (D).**
   - RQ-D1: How does disagreement among reviewers affect acceptance?
   - RQ-D2: In borderline cases, how do reviewer confidence and sentiment influence acceptance?
2. **Review Interaction Logs (I).**
   - RQ-I1: Do rebuttal response speed and length affect acceptance?
   - RQ-I2: Are more exchanges and greater depth signals of rescue or risk?
3. **Submission Strategies & External Metadata (S).**
   - RQ-S1: How does submission timing (early vs. last-minute) affect final acceptance?
   - RQ-S2: How does arXiv posting influence acceptance?
   - RQ-S3: Does providing reproducibility information affect final decisions?
4. **Paper Structure & Language Quality (P).**
   - RQ-P1: How do visual elements (tables, figures, equations) relate to novelty and contribution?
   - RQ-P2: How do English fluency and clarity relate to acceptance rates?
   - RQ-P3: How does reference recency influence novelty and reviewer evaluation?

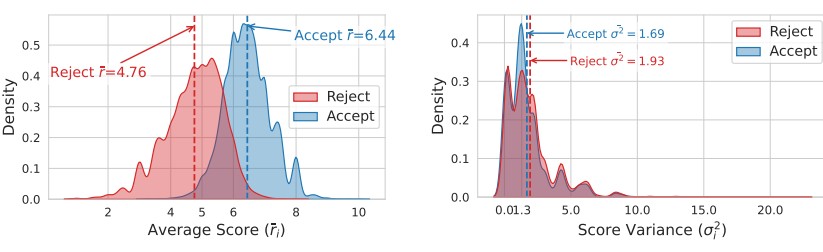

(a) Distribution of scores         (b) Distribution of score variance

Figure 1: Distribution of scores ($\bar{r}_i$) and score variance ($\sigma_i^2$)

**Dataset.** Our analysis uses the complete ICLR 2017–2025 OpenReview corpus of 28,358 submissions. It includes reviews, rebuttals, discussions, meta-reviews, final decisions, arXiv metadata, and PDF-derived features (tables, figures, equations, readability, reference recency). This dataset enables a process-centric, multi-dimensional analysis of peer review.

Detailed information on dataset selection and construction is provided in **Appendix B**, and the dataset will be released to the research community for future studies. In addition, the results of all statistical significance tests for our experiments are reported in **Appendix C**, including correlation tests like Spearman correlation (Spearman, 1961), Pearson correlation (Pearson, 1895), and mean-difference tests like Mann–Whitney U test (Nachar et al., 2008), Welch's $t$-test (Welch, 1947), and regression-based analyses like logistic regression (Hosmer Jr et al., 2013)).

## 3   REVIEWER DISAGREEMENT & MEDIATION: COLLECTIVE DECISION-MAKING PATTERNS

The intrinsic quality of a paper is central in academic evaluation, yet *reviewer disagreement* is inevitable when reviewers with different backgrounds and standards assess the same work. Platforms such as OpenReview disclose scores, comments, confidence ratings, meta-reviews, and interaction logs (discussion threads, rebuttals, score revisions), enabling quantitative analysis of collective decision-making previously inaccessible. In this chapter, we investigate how reviewer disagreement relates to paper acceptance. We focus on three key dimensions: **score variance**, **reviewer confidence**, and the **sentiment of review comments**.

**Experimental Design.** We used ICLR 2017–2025 OpenReview dataset. The analysis unit was defined as papers with at least two reviewers providing ratings, confidence, and final decision. Data from 2020 were excluded due to missing confidence scores. The final dataset includes 23,444 papers.

Notation is defined as follows. Let $d_i$ denote the $i$-th paper, $r_i^k$ the score assigned by reviewer $k$ for paper $d_i$, $c_i^k$ the confidence of reviewer $k$, and $y_i \in \{0, 1\}$ the final decision (1 for accept, 0 for reject). Using these, we construct: the mean score $\bar{r}_i = \frac{1}{K_i} \sum_{k=1}^{K_i} r_i^k$, the score variance $\sigma_i^2 = \frac{1}{K_i} \sum_{k=1}^{K_i} (r_i^k - \bar{r}_i)^2$, the mean confidence $\bar{c}_i = \frac{1}{K_i} \sum_{k=1}^{K_i} c_i^k$, and the confidence variance $\sigma_{c,i}^2 = \frac{1}{K_i} \sum_{k=1}^{K_i} (c_i^k - \bar{c}_i)^2$, where $K_i$ is the number of reviewers for paper $d_i$.

### 3.1   RQ-D1: HOW DOES DISAGREEMENT AMONG REVIEWERS AFFECT ACCEPTANCE?

Reviewer score disagreements were *consistently* observed across ICLR 2017–2025. As shown in Figure 1, both accepted and rejected papers exhibited non-negligible score variance ($\sigma_i^2$). The mean score of accepted papers was about 6.44, with accept and reject decisions clustering in the 5-6 range.

Based on the official ICLR review score guidelines, we divide papers into three groups: High-Score ($\bar{r}_i > 6$), Borderline ($5 \leq \bar{r}_i \leq 6$), and Low-Score ($\bar{r}_i < 5$). Detailed scale definitions and semantic interpretations are provided in Appendix B.3.

Figure 2 compares acceptance rates by score variance across groups. In the Low-Score group, accepted papers showed *higher variance* than rejected papers. This suggests that even with a low average score, a few high ratings can increase variance and modestly raise acceptance probability. In the High-Score group, rejected papers exhibited greater variance, indicating that a single strong negative review could override otherwise high scores.

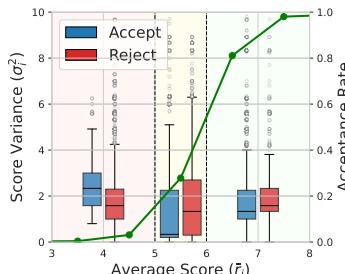

In the Borderline group, accepted papers had lower median variance than rejected ones, showing that reviewer agreement improved acceptance odds when scores were near the decision threshold.

See Table 3 in Appendix C.1, the Low-Score group exhibited a significant positive correlation. This statistically confirms that **reviewer disagreement, when average scores are low, can sometimes lead to acceptance** rather than rejection.

Figure 2: Variance and acceptance by score group and decision

### 3.2 RQ-D2: IN BORDERLINE CASES, HOW DO REVIEWER CONFIDENCE AND SENTIMENT INFLUENCE ACCEPTANCE?

When reviewer disagreement is pronounced—particularly in the Borderline group ($5 \leq \bar{r}_i \leq 6$), where the final outcome is uncertain—score variance alone is insufficient to explain acceptance. In this context, our study seeks to identify which reviewer attributes influence the final decision.

We focus on two key factors. The first is the level of reviewer *confidence*, both at the individual level ($c_i^k$) and at the paper level (mean confidence $\bar{c}_i$). The second is the *sentiment* of review comments, specifically whether they are predominantly positive, neutral, or negative.

**Confidence Analysis.** Since reviewers have different expertise areas, their confidence scores naturally vary. As shown in Figure 3, the average reviewer confidence on a 1–5 scale was about 3.60, with negligible differences between accepted papers (3.58) and rejected papers (3.61). Confidence variance also showed nearly *identical* distributions across the two groups. These results, confirmed in Appendix C.2, indicate that confidence does not have a *meaningful* correlation with acceptance outcomes in the Borderline group.

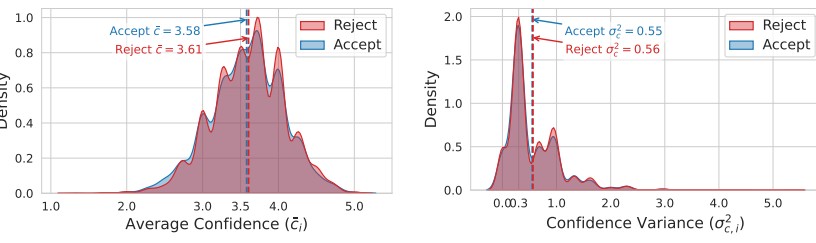

(a) Distribution of $\bar{r}_i$ in borderline cases  (b) Distribution of $\sigma_i^2$ in borderline cases

Figure 3: Distribution of reviewer confidence in borderline cases

**Sentiment Analysis.** We examine how the sentiment distribution of review comments affects acceptance among Borderline papers ($5 \leq \bar{r}_i \leq 6$), as illustrated in Figure 4. Formally, we define the following units and notations. Let $m_i^{k,j}$ denote the $j$-th comment of reviewer $k$ on paper $d_i$. Each comment $m_i^{k,j}$ is mapped by a pretrained multilingual sentiment model[1] into a probability vector over positive / neutral / negative class. The dominant sentiment of a review $\hat{s}_i^k$ is determined by majority vote across its comments, and the paper-level sentiment $\hat{s}_i$ is the most frequent across reviewers (ties resolved by average intensity).

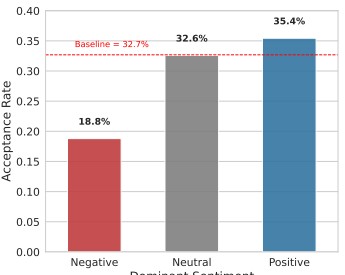

Figure 4: Acceptance rates by dominant sentiment in borderline

The overall acceptance rate of Borderline papers was 32.7%. By dominant sentiment $\hat{s}_i$, acceptance was 35.4% for Positive, 32.6% for Neutral, and 18.8% for Negative.

This indicates that **sentiment is a strong** signal in meta-review deliberations: negative comments strongly constrain acceptance, while positive ones provide a modest boost.

---

[1]https://huggingface.co/nlptown/bert-base-multilingual-uncased-sentiment

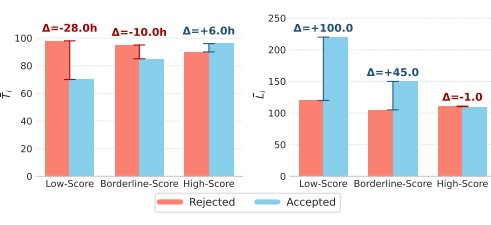 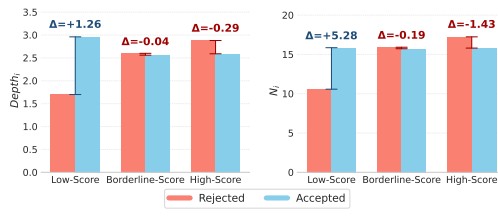

(a) Response speed & length                    (b) Interaction depth & Response Count

Figure 5: Rebuttal dynamics. (a) Low/Borderline: faster and longer replies ↑ acceptance; High: no clear effect. (b) Depth ($Depth_i$) and count ($N_i$): helpful in Low, but neutral/negative when excessive in Borderline/High.

### 3.3 Summary and Implications

Our findings can be summarized as follows: (1) Reviewer disagreement generally lowers acceptance probability, with the effect strongest in the High-Score and Borderline groups where variance directly reduces acceptance chances. (2) In the Low-Score group, *disagreement sometimes helps*: a few positive reviews increase variance and can raise acceptance probability. (3) Reviewer confidence does not significantly predict acceptance in Borderline cases. (4) Sentiment is a decisive factor: negative comments sharply reduce acceptance, while positive comments slightly improve it. Overall, these results show that disagreement is not mere noise but an important mechanism in peer review decision-making. They highlight the need for meta-reviewers to balance quantitative signals (score variance) with qualitative cues (review sentiment) when mediating reviewer disagreements.

## 4 Review Interaction Logs: Understanding Interactive Dynamics

A defining feature of modern peer review platforms such as OpenReview is that the **dialogue** between authors and reviewers during rebuttal is fully logged and public. These records form a basis for meta-reviewer decisions, so the timing, volume, depth, and frequency of exchanges can affect the final outcome. In this chapter, we analyze ICLR 2017-2025 data to examine how rebuttal responsiveness (speed and length, RQ-I1) and interactional dynamics (depth and response count, RQ-I2) influence acceptance.

**Experimental Design.** Our analysis targets 20,666 papers (73.63%) from the ICLR 2017-2025 corpus, selected from a total of 28,383 submissions. These papers include complete records of reviews, author rebuttals, meta-reviews, and final decisions, and contain at least one response (latency $\leq$ 336H). Interactions outside the official rebuttal window (typically 7-14 days) were excluded.

Formally, we define the following units and notations. Let $m_i^{k,j}$ denote the $j$-th comment of reviewer $k$ on paper $d_i$. The mean response length across all responses in paper $d_i$ is denoted by $\bar{L}_i$, and $\bar{T}_i$ represents the mean latency of author responses in $d_i$. We denote by $Depth_i$ the maximum depth of reviewer–author dialogue trees in paper $d_i$, and by $N_i$ the total number of author responses in $d_i$.

Descriptive statistics across score groups (acceptance rate, mean $\bar{T}_i$, mean $\bar{L}_i$) are reported in Appendix C.3 (Table 5). Differences between groups are statistically significant.

### 4.1 RQ-I1: Do Rebuttal Response Speed and Length Affect Acceptance?

Most conferences allow 7-14 days for rebuttal submissions. Within this limited period, the promptness and thoroughness of an author's response become key signals of persuasiveness to reviewers and meta-reviewers.

As shown in Figure 5a, in Low-Score and Borderline groups, faster responses ($\bar{T}_i \downarrow$) and longer responses ($\bar{L}_i \uparrow$) are **significantly associated with higher acceptance rates**. In contrast, for the High-Score group, neither metric exhibits a significant effect.

Statistical tests (Appendix C.3) show that response speed and length significantly affect acceptance for Low- and Borderline-Score papers, but not for High-Score papers.

## 4.2 RQ-I2: Are more exchanges and greater depth signals of rescue or risk?

While authors strive to maximize their rebuttal efforts, excessive back-and-forth can burden reviewers and trigger negative reactions. To test this, we examine the depth of interaction chains ($Depth_i$) and the total number of author responses ($N_i$).

As shown in Figure 5b: While authors strive to maximize their rebuttal efforts, excessive iterations can burden reviewers and trigger negative reactions. To test this, we examine the depth of interaction chains ($Depth_i$) and the total number of author responses ($N_i$).

This suggests that intensive interaction is beneficial only at the lower end of the score spectrum, whereas in higher tiers it risks being interpreted as argumentative or inefficient.

**Effective range of interaction.** we analyzed acceptance rates across binned intervals of response counts and interaction depths (see Appendix C.4).

The findings reveal that for the Low-Score group ($\bar{r}_i < 5$), more responses consistently increase acceptance rates, demonstrating the effectiveness of rebuttals as a salvage strategy. For the Borderline group ($5 \leq \bar{r}_i \leq 6$), more responses yield modest gains, but excessive depth can backfire. Finally, for the High-Score group ($\bar{r}_i > 6$), greater response counts and deeper discussions correlate with declining acceptance rates, suggesting that prolonged debate may undermine strong papers. Overall, the analysis identifies an optimal interaction window: **dialogue depth of 2–3 ranges and a moderate number of responses tend to maximize acceptance prospects**, beyond which diminishing or negative returns emerge.

### 4.3 Summary and Implications

This chapter yields three insights. First, faster and longer rebuttals aid Low- and Borderline-Score papers but have little effect on High-Score ones. Second, interaction depth and frequency diverge: extensive exchanges help Low-Score cases, moderate discussion benefits Borderline ones, and greater dialogue harms High-Score papers. Third, there is an "effective range of interaction": some rebuttal activity helps, but overly prolonged or deep debates reduce acceptance chances.

These findings show the rebuttal stage as a decisive arena that can alter a paper's trajectory. Authors must tailor strategies to their score group, while meta-reviewers should weigh not only quantitative indicators but also their contextual meaning.

## 5 Submission Strategies & External Metadata: Signals Beyond the Manuscript

In modern conference peer review, decisions reflect not only manuscript quality but also *external signals* such as submission timing, public posting, and code or data availability. At ICLR, using OpenReview, these signals are explicit, offering insights into double-blind review. This chapter analyzes how submission strategies and external metadata affect acceptance.

**Experimental Design.** Our analysis uses the ICLR 2017–2025 OpenReview dataset. From 28,086 submissions, we include papers with a recorded final decision and submission timing relative to the deadline. We address three research questions.

### 5.1 RQ-S1: How does submission timing (early vs. last-minute) affect final acceptance?

Submission timing is defined as the number of days remaining until the deadline. As shown in Figure 6a, early submissions ($\geq$ 7 days before deadline) had an acceptance rate of about 44.0%, much higher than the overall mean of 38.9%, and even slightly higher than last-minute submissions (about 43%). Similarly, Figure 6b indicates that early submissions also achieved higher average reviewer scores. This suggests that timing itself can influence evaluation outcomes.

In Figure 6c, The Gini Index of reviewer disagreement reveals that last-minute submissions experienced higher disagreement, likely due to incomplete preparation or weaker clarity. In contrast, early submissions exhibited lower variance, suggesting more consistent evaluations across reviewers.

Statistical results are deferred to Appendix C.5. Early submissions show higher acceptance, stronger scores, and less disagreement.

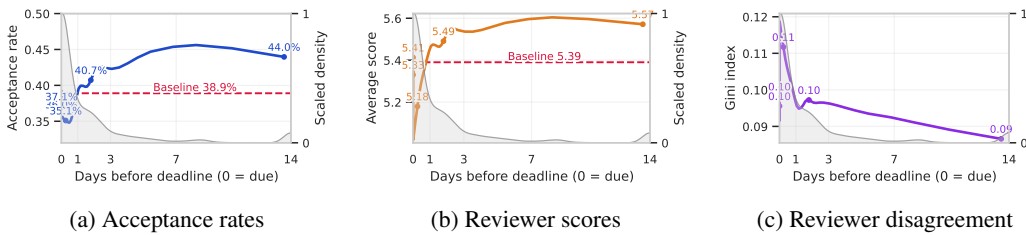

(a) Acceptance rates      (b) Reviewer scores      (c) Reviewer disagreement

Figure 6: Effects of submission timing on acceptance. The gray area shows the density distribution of submissions, while the colored lines represent acceptance rate (blue), average score (orange), and Gini index (purple).

## 5.2 RQ-S2: HOW DOES ARXIV POSTING INFLUENCE ACCEPTANCE?

We next analyze arXiv postings, focusing on 14,012 papers matched by title and author overlap ($\geq 50\%$). Matching was determined by (1) BLEU score $\geq 0.8$ between dataset and arXiv titles, and (2) author overlap exceeding 50%.

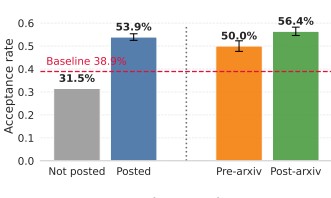

(a) arXiv posting

Figure 7a compares acceptance rates across posting conditions. Papers on arXiv *consistently* outperformed non-posted papers. Interestingly, timing also mattered: papers posted simultaneously with or after post submission ($\leq 0$h) showed about 6.4%p higher acceptance than preprints posted earlier.

This finding suggests that beyond signaling openness, the coordination of arXiv release with submission acts as a indicator of readiness and completeness. Thus, arXiv posting increases acceptance likelihood, and strategically timed posting amplifies its effect.

Finally, we examine whether providing **R**eproducibility-**S**upporting Resources ($RS$) affects outcomes. Using the supplementary material, code, and data fields, we categorized submissions into with-$RS$ and without-$RS$ groups.

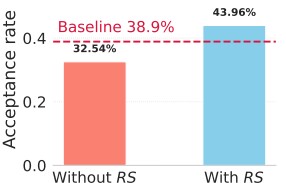

(b) Reproducibility Supporting Resources

Figure 7: (a) Acceptance rates by arXiv posting status/timing; (b) Acceptance w/, w/o $RS$ (code/data).

As shown in Figure 7b, papers with-$RS$ had an acceptance rate of 43.96%, significantly above the average (38.9%) and *much higher* than papers without-$RS$ (32.54%). Detailed statistical tests are provided in Appendix C.6, confirming that RS provision strongly predicts higher acceptance probability and functions as a decisive trust signal for reviewers.

## 5.3 SUMMARY AND IMPLICATIONS

This chapter yields three main insights. First, submission timing strongly affects outcomes: early submissions achieve higher acceptance rates, higher reviewer scores, and lower disagreement. Second, arXiv posting substantially increases acceptance likelihood, with simultaneous or post-submission posting serving as a stronger signal than early preprints (Figure 7a). Third, reproducibility resources (code/data) significantly boost acceptance, both statistically and practically, reinforcing trust in the paper's credibility (see Figure 7b).

Taken together, submission strategies and external metadata are not peripheral factors but central determinants of acceptance. Even under double-blind review, external signals are not ignored. For authors, managing submission timing, aligning arXiv posting, and providing reproducibility resources serve as effective levers to increase acceptance probability.

## 6 THE IMPACT OF VISUAL, LINGUISTIC, AND REFERENCE PATTERNS ON EVALUATION

The novelty, contribution, and experimental results of a paper are the primary factors reviewers consider when evaluating its quality. However, external aspects such as style, structure, layout, and language clarity may also influence reviewer decisions. This chapter quantitatively analyzes how non-core factors—visual elements (e.g., tables, figures), linguistic clarity, and reference recency—affect evaluations of novelty and contribution.

### 6.1 RQ-P1: How do visual elements (tables, figures, equations) relate to novelty and contribution?

Visual devices (e.g., tables, figures, equations) help readers grasp contributions and are often used to summarize findings. This section examines how their counts correlate with novelty and contribution scores. Our dataset includes 6,473 ICLR 2022–2023 papers with novelty scores and 8,624 ICLR 2025 papers with contribution scores. Visual elements were extracted from PDFs using pdfplumber[2], with each type capped at ten to mitigate parsing errors.

Appendix C.7 reports that visual elements are *significantly* associated with novelty and contribution: tables and equations enhance empirical contributions, whereas excessive visuals may lower technical novelty. In summary, technical novelty is rated higher when fewer tables and figures are presented, whereas empirical novelty is strengthened by including more tables and figures. Contribution scores also increase when more tables and equations are included. Thus, concise visual design may be effective when the goal is to highlight the originality of an idea, while abundant tables and equations are advantageous for emphasizing empirical persuasiveness or contribution.

### 6.2 RQ-P2: How do English fluency and clarity relate to acceptance rates?

Readable and fluent writing can leave a positive impression on reviewers. In this section, we analyze papers across various linguistic dimensions (fluency and readability) and examine how these metrics relate to acceptance rates across both the pre- and post-LLM eras.

**Metrics.** We consider three linguistic dimensions: vocabulary diversity (unique-to-total word ratio); average sentence length; average words per sentence; and readability indices (Flesch Reading Ease (Flesch, 1943), Flesch–Kincaid Grade (Kincaid et al., 1975), Gunning Fog Index (Gunning, 1952)).

**Dataset Split.** We define the introduction of LLMs into writing workflows with the release of ChatGPT (Nov. 30, 2022). Since the ICLR 2023 deadline (Sept. 11, 2022) predates this, papers from 2024 onward are considered Post-LLM. Thus, we classify submissions into two groups: Pre-LLM (ICLR 2017–2023) and Post-LLM (ICLR 2024–2025).

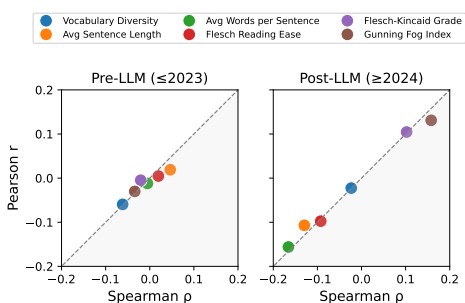

Figure 8: Correlation between metrics and acceptance rates in the pre- and post-LLM eras

Figure 8 illustrates the relationship between linguistic metrics and acceptance rates before and after the introduction of LLMs. Points lying along the diagonal indicate that Spearman's rank correlation and Pearson's correlation yield the same direction of association, while greater distance from the center reflects stronger absolute correlation values, i.e., a larger influence on acceptance rates. Detailed statistical results are in Appendix C.8. In the pre-LLM era (2017–2023), **easier-to-read writing and slightly longer sentences** were modestly advantageous, though overall effects were weak. By contrast, in the post-LLM era (2024–2025), these patterns reversed and became stronger: professional, somewhat difficult tone was favored, while short and concise sentences emerged as a clear positive signal, reflecting the principle of **"Professional but concise."**

### 6.3 RQ-P3: How does reference recency influence novelty and reviewer evaluation?

Reviewers often assess novelty by considering how well a paper engages with the *latest* research. We quantify reference recency using three metrics: the average reference age (submission year minus reference year), the recent references ratio (fraction of references published within two years), and the old references ratio (fraction of references older than five years).

As you can see with Table 13 in Appendix C.9, older references—as captured by the average reference age—strongly decrease empirical novelty, while their effects on technical novelty are negligible. A higher recent references ratio significantly boosts empirical novelty, whereas a higher old references ratio significantly reduces it. Figure 11 in Appendix C.9 confirms these findings.

In conclusion, papers that sufficiently reflect the latest research context receive higher empirical novelty ratings. Technical novelty, however, is largely determined by the originality of ideas or formalization and is not meaningfully influenced by reference recency.

---

[2]https://github.com/jsvine/pdfplumber

## 6.4 SUMMARY AND IMPLICATIONS

This chapter yields three main insights. First, visual elements show divergent effects: tables and equations enhance novelty and contribution, while excessive visuals may reduce technical novelty. Second, the notion of "good writing" has shifted: before LLMs, **clarity and readability** improved acceptance, but after LLMs, **a professional, concise tone** received higher ratings. Third, reference recency is key: younger references and higher ratios of recent citations significantly increase empirical novelty. Together, these findings show that style, language, and references are substantive signals shaping reviewer evaluations.

## 7 DISCUSSION

This study uses process-level data from modern peer review to identify determinants of acceptance across four dimensions: process, interaction, signaling, and style. OpenReview's transparency enables moving beyond scores to examine how decisions form through disagreement, dialogue, external signals, and stylistic practices.

**Integrated interpretation.** Reviewer disagreement reduces acceptance in High/Borderline cases but can help Low-score papers with a few favorable reviews. Fast, sufficiently long rebuttals act as a rescue strategy, with optimal depth at 2–3 exchanges. External signals—early submission, synchronized/post-submission arXiv posting, and reproducibility resources—increase acceptance odds. Style also matters: tables/equations aid novelty and contribution, while post-LLM reviewers favor concise professional writing and recent references.

**Peer review as a signaling game.** Peer review functions as a signaling game of belief updates: disagreement reflects information divergence, interaction logs serve as costly signals of effort, submission timing and RS indicate readiness, and stylistic choices shape cognitive load and credibility.

**Guidelines for authors.** For Low-Score papers ($<5$), respond quickly and substantively, keep dialogue depth at 2–3, provide RS, and write concisely with tables/equations. For Borderline papers (5–6), balance speed and substance, counter negative sentiment with precise clarifications, leverage early submission and synchronized arXiv posting, and emphasize novelty with recent references. For High-Score papers ($>6$), avoid excessive interactions and instead consolidate contributions with RS and concise conclusions.

**Recommendations for meta-reviewers and organizers.** Meta-reviewers should apply variance checks to prevent outlier dominance and manage interactions by encouraging 2–3 dialogue turns while treating longer debates with caution. They should separate evidence from tone to buffer undue negativity, and explicitly reward reproducibility resources by linking credits to validation. Finally, concise professional style should be valued, but substance must take priority over polish.

**Shifts in the LLM era.** After 2024, readability lost its advantage, replaced by a preference for professional, concise writing. This shift stems from LLMs raising baseline quality and reviewers valuing brevity under time constraints. Institutions should separate style from substance, and authors should deliver dense, evidence-backed claims in minimal sentences.

**Limitations and future work.** Findings are based on ICLR 2017–2025, limiting generalizability, and results are correlational rather than causal. Data extraction and sentiment analysis introduce noise, and offline/private interactions remain unobserved. Future studies should compare across disciplines, use agent-based simulations of interaction rules, and analyze internal argument–evidence structures.

## 8 CONCLUSION

This study uncovers how process-level data shapes outcomes: the structure of disagreement, optimal interaction windows, external reproducibility and disclosure signals, and the alignment of concise professional writing with up-to-date references. These insights provide authors with strategic guidance and give meta-reviewers and organizers a foundation for fairer, reproducible practices. Moving forward, causal testing and policy experiments will be essential. Ultimately, when good ideas meet good processes, peer review can achieve more trustworthy and equitable consensus.

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

## A    RELATED WORK

**Traditional Peer Review and Its Limitations.**    Traditional peer review in journals and conferences has relied on a closed and blind system (Wicherts, 2016; Peh, 2022), where only the final decision (accept/reject) was disclosed. This structure lacked transparency (Karhulahti & Backe, 2021; Tennant & Ross-Hellauer, 2020), made it difficult to systematically evaluate review quality (Wicherts, 2016), and provided no interactional data between authors and reviewers (Chong & Lin, 2024; Ross-Hellauer et al., 2017). As a result, earlier studies were constrained to analyzing limited information such as reviewer scores or final outcomes (Tennant & Ross-Hellauer, 2020).

**The Transformation of Peer Review in the Modern Era.**    With the advent of platforms such as OpenReview (Yang, 2025), conferences like ICLR began to disclose the entire review process, from submission to final decision (OpenReview.net, 2017). This shift made reviewer scores, comments, rebuttals, and meta-reviews all available as analyzable data (Yang, 2025; Huang et al., 2023), thereby expanding the dimensions of peer review research. Crucially, interaction records between authors and reviewers (Chong & Lin, 2024; Huang et al., 2023), information about code and data availability (Kang et al., 2023) have been systematically archived. This has enabled the field to move beyond an outcome-centric perspective toward a process-centric analysis of peer review dynamics (Aczel et al., 2025).

**Existing Studies on What Constitutes a "Good Paper" and Predicts Acceptance.**    Prior research has attempted to define what makes a "good paper" in terms of citations (Pottier et al., 2024), and reproducibility (Tennant & Ross-Hellauer, 2020). Studies on linguistic factors examined how elements such as the title (Pottier et al., 2024), abstract (Pho, 2008), textual clarity, and difficulty (Priyadarshini et al., 2023) influenced acceptance outcomes. Other work investigated the role of visual and structural features—tables (Divecha et al., 2023), figures (Ariga & Tashiro, 2022), and equations—in shaping reviewer evaluations. Metadata such as code/data availability (Kang et al., 2023), arXiv posting, and citation counts have also been explored, alongside process-related features such as reviewer reliability, and rebuttal effectiveness (Huang et al., 2023). However, most prior studies were limited in scope: many focused on a single year (Jen et al., 2018), a small subset of data, or a narrow set of explanatory factors (Aczel et al., 2025).

**Research Gap and the Need for Process-Centric Approaches.**    Existing literature has largely emphasized outcome-based analyses (Zhang et al., 2022; Gallo et al., 2014)—acceptance versus rejection, or citation impact(Kho et al., 2020). Yet in modern peer review environments, especially those built on OpenReview (Zhang et al., 2025), abundant process-level data is available, enriched by external metadata such as arXiv postings or reproducibility resources. This creates a pressing need for research frameworks that integrate both process signals and external signals in explaining paper acceptance. Our study addresses this gap by systematically analyzing ICLR 2017–2025 data, providing empirical evidence for shifting peer review research from an outcome-centric to a process-centric paradigm.

## B    DATASET DETAILS

### B.1    DATASET SELECTION

We reviewed several candidate datasets for modern peer review research. While NeurIPS and EMNLP (since 2019) and ACL (since 2020 through ARR) have partially adopted OpenReview, many of their records lack meta-reviews or final decisions, limiting their completeness. PeerRead (2013–2017) provides early peer review data but does not include rebuttals or meta-reviews.

By contrast, ICLR 2017–2025 offers complete and consistent coverage: reviewer scores, confidence levels, textual comments, author rebuttals, reviewer–author discussions, meta-reviews, and final decisions. The dataset contains 28,358 submissions, with annual acceptance rates in the low 30–40% range. Submissions grew from 490 in 2017 to 8,629 in 2025, a 17-fold increase.

### B.2    DATASET CONSTRUCTION

We collected the data through the OpenReview API (v1 and v2), excluding withdrawn or incomplete cases. In addition, we retrieved arXiv metadata (posting status, upload date, versioning) and aligned

it with submission timestamps. For paper content, we downloaded PDFs and parsed them using pdfplumber, extracting both textual and visual features.

The following features were measured:

- Visual density: counts of tables, figures, and equations.
- Language quality: readability indices (Flesch, FK Grade, Gunning Fog), average sentence length, lexical diversity, and passive voice rate.
- Reference recency: average reference age, ratio of recent references, and ratio of old references.

The final dataset integrates:

- Paper content (PDF text and structure)
- Reviews and scores
- Rebuttals and interaction logs
- Meta-reviews and decisions
- ArXiv metadata

This dataset enables a process-centric, multi-dimensional analysis of peer review alongside structural and stylistic features of manuscripts. Following anonymization and copyright checks, we will release the dataset to the research community.

### B.3 SCORE GROUPS AND SEMANTIC SCALE

Table 1 reports the yearly submission outcomes for ICLR 2017–2025. The data show a dramatic increase in total submissions, rising from only 490 papers in 2017 to 8,629 in 2025, a more than 17-fold growth over nine years. This rapid expansion reflects both the surge of research activity in deep learning and the increasing visibility and prestige of ICLR within the AI/ML community.

Despite this large growth in volume, the overall acceptance ratio has remained relatively stable in the 30–40% range across years. This indicates that the program committee has scaled the review process effectively, maintaining a consistent level of selectivity even as the conference size grew. Such stability ensures that acceptance at ICLR remains a strong quality signal, regardless of year-to-year fluctuations in submission counts.

From a data perspective, this long-term growth highlights both the scalability of the dataset and the robustness of acceptance thresholds. It also provides a unique opportunity for process-level analysis across heterogeneous submission volumes, enabling us to investigate whether the determinants of acceptance vary with conference size or remain consistent across different stages of ICLR's evolution.

Table 1: Submission outcomes by year (ICLR 2017–2025)

| Year | Accept | Reject | Total |
|------|--------|--------|-------|
| 2017 | 198 | 292 | 490 |
| 2018 | 336 | 575 | 911 |
| 2019 | 502 | 917 | 1,419 |
| 2020 | 687 | 1,526 | 2,213 |
| 2021 | 859 | 1,735 | 2,594 |
| 2022 | 1,094 | 1,524 | 2,618 |
| 2023 | 1,573 | 2,223 | 3,796 |
| 2024 | 2,263 | 3,450 | 5,713 |
| 2025 | 3,704 | 4,925 | 8,629 |

Based on the official ICLR review score guidelines, we divide papers into three groups according to their mean review score $\bar{r}_i$:

- High-Score Group: $\bar{r}_i > 6$
- Borderline Group: $5 \leq \bar{r}_i \leq 6$
- Low-Score Group: $\bar{r}_i < 5$

Table 2 provides the official score scale and its semantic interpretations. This classification is used consistently throughout our analysis to examine acceptance determinants across different score regimes.

Table 2: Official ICLR Review Score Scale and Semantic Interpretations

| Group | Score | Semantic Comment |
|---|---|---|
| Low | 1 | Reject, strong reject |
| | 2 | Strong rejection |
| | 3 | Weak Reject, reject, not good enough, Clear rejection |
| | 4 | Ok but not good enough – rejection |
| Borderline | 5 | Marginally below the acceptance threshold |
| | 6 | Marginally above the acceptance threshold, Weak Accept |
| High | 7 | Good paper, accept |
| | 8 | Top 50% of accepted papers, clear accept, accept, good paper |
| | 9 | Top 15% of accepted papers, strong accept |
| | 10 | Strong accept, should be highlighted at the conference, Top 5% of accepted papers, seminal paper |

## B.4 DATASET JSON SCHEMA

For reproducibility, we provide the JSON schema of our refined dataset structure. The raw API data from OpenReview is not standardized and thus difficult to use directly, with field formats varying across years and venues. We refined and unified these records into a structured JSON schema that ensures consistency and facilitates downstream analysis. This curated dataset will be released publicly to support future research on peer review.

An illustrative excerpt is shown below:

```
{
  "ICLR_2017": {
    "Identity Matters in Deep Learning": {
      "venue": ["str"],
      "venueid": ["str"],
      "note_ids": ["str"],
      "keywords": [],
      "abstract": ["str"],
      "chair_decisions": ["str"],
      "reviewer_comments": {
        "AnonReviewer3": {
          "comment": "str",
          "rating": "str",
          "confidence": "str"
        },
        "AnonReviewer1": {
          "comment": "str",
          "rating": "str",
          "confidence": "str"
        },
        "AnonReviewer2": {
          "comment": "str",
          "rating": "str",
          "confidence": "str"
        }
      },
      "metareviewer_comments": {
        "comment": "str"
      },
      "metadata_use": "bool",
      "final_decision": "str"
    },
    "Differentiable Canonical Correlation Analysis": {
      ...
    }
  }
}
```

This schema defines the unified format of our dataset. It consolidates heterogeneous records from different years into a consistent structure. Through this representation, users can reliably access re-

views, meta-reviews, and associated metadata. The schema thus ensures comparability and usability across the entire ICLR 2017–2025 corpus.

## C  PER-RQ EXPERIMENTAL RESULTS AND STATISTICAL SIGNIFICANCE

In this study, we employed multiple statistical tests to examine the robustness and reliability of our findings. The following measures were primarily used:

- Spearman Correlation ($\rho$): Non-parametric rank correlation test to measure the monotonic relationship between variables.
- Pearson Correlation ($r$): Linear correlation coefficient for continuous variables, assuming approximate normality.
- Mann–Whitney U Test: A non-parametric test for assessing whether two independent samples come from the same distribution, often used as an alternative to the two-sample t-test when normality cannot be assumed.
- Welch's t-test: A robust version of the two-sample t-test that does not assume equal population variances.
- Logistic Regression Coefficient ($\beta$) and Odds Ratio (OR): Regression-based analysis to estimate the likelihood of acceptance as a function of predictors, providing both effect size ($\beta$) and interpretability (OR).

Together, these tests provide a balanced framework for analyzing correlations, mean differences, and predictive effects, ensuring both statistical validity and practical interpretability across heterogeneous data.

### C.1  RQ-D1: HOW DOES DISAGREEMENT AMONG REVIEWERS AFFECT ACCEPTANCE?

**Visualization of Reviewer Disagreement.**  Figure 9 further illustrates this relationship. Acceptance rates declined as variance increased, especially in the High-Score group, where acceptance dropped sharply (–0.28 at the 99th percentile). In contrast, Low-Score papers showed a mild acceptance increase when variance was small, before declining again.

This indicates that disagreement can sometimes benefit low-scoring papers.

**Statistical Tests on Reviewer Disagreement.**  This indicates that disagreement can sometimes benefit low-scoring papers. As shown in Table 3, correlation tests confirm that score variance is negatively correlated with acceptance in the High-Score and Borderline groups, but positively in the Low-Score group. Thus, disagreement tends to hurt acceptance when average scores are high or borderline, but can help low-scoring papers if at least one reviewer is supportive.

Table 3: Correlation tests between score variance and acceptance (RQ-D1)

| Group | N | Spearman | Pearson | Mann–Whitney U | Welch t | Logit $\beta$/OR |
|---|---|---|---|---|---|---|
| Overall | 23,444 | -0.074 (7.13e-30) | -0.067 (1.85e-24) | 60176515.000 (8.48e-30) | -10.417 (2.37e-25) | -0.08 / 0.923 (3.84e-24) |
| Low-Score-Group | 7,255 | 0.086 (2.52e-13) | 0.070 (1.94e-09) | 581060.500 (2.77e-13) | 5.557 (1.70e-07) | 0.206 / 1.228 (9.71e-09) |
| Border-Score-Group | 9,549 | -0.105 (1.26e-24) | -0.033 (1.10e-03) | 8745381.000 (1.67e-24) | -3.217 (1.30e-03) | -0.036 / 0.965 (0.00112) |
| High-Score-Group | 6,640 | -0.141 (5.15e-31) | -0.131 (6.26e-27) | 1098371.000 (9.99e-31) | -8.301 (7.73e-16) | -0.237 / 0.789 (5.41e-24) |

### C.2  RQ-D2: IN BORDERLINE CASES, HOW DO REVIEWER CONFIDENCE AND SENTIMENT INFLUENCE ACCEPTANCE?

**Statistical Tests on Reviewer Confidence.**  Table 4 reports the statistical tests on reviewer confidence in Borderline cases ($5 \leq \bar{r}_i \leq 6$). As the results show, average confidence and confidence variance exhibited negligible differences between accepted and rejected papers, and the statistical tests confirm that reviewer confidence does not meaningfully correlate with acceptance outcomes.

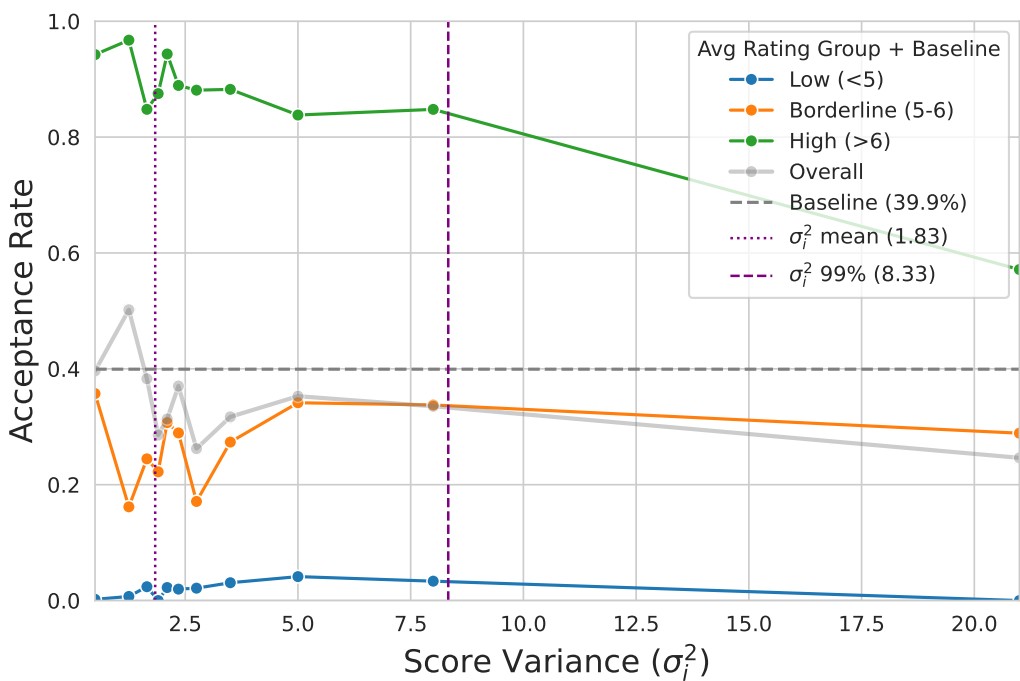

Figure 9: Acceptance rate trends across score variance percentiles (RQ-D1)

Table 4: Statistical tests on reviewer confidence in Borderline cases (RQ-D2)

| Group | N | Spearman | Pearson | Mann–Whitney U | Welch t | Logit $\beta$/OR |
|---|---|---|---|---|---|---|
| Borderline (5–6) | 9,549 | -0.022 (3.08e-02) | -0.025 (1.57e-02) | 9759663.000 (3.08e-02) | -2.406 (1.61e-02) | -0.111 / 0.895 (1.57e-02) |

C.3  RQ-I1: DO REBUTTAL RESPONSE SPEED AND LENGTH AFFECT ACCEPTANCE?

**Descriptive Statistics of Rebuttal Responses.**  Table 5 summarizes descriptive statistics across score groups (acceptance rate, mean $\bar{T}_i$, mean $\bar{L}_i$). Interestingly, Low-Score papers tend to produce longer and more delayed responses, while High-Score papers exhibit shorter responses despite their high acceptance rate.

Table 5: Descriptive statistics of rebuttal responses across score groups (RQ-I1)

| Group | N | Accept Rate | $\bar{T}_i$ | $\bar{L}_i$ |
|---|---|---|---|---|
| Overall | 20,666 | 0.39 | 91.8 | 127.1* |
| Low-Score-Group | 6,599 | 0.07 | 94.2 | 170.0* |
| Border-Score-Group | 8,710 | 0.39 | 90.4 | 128.0* |
| High-Score-Group | 5,357 | 0.93 | 91.1 | 110.0* |

**RQ-I1: Statistical Tests on Rebuttal Response Speed and Length.**  Statistical tests in Table 6 and 7 confirm this pattern: response speed and length are significant predictors of acceptance for Low- and Borderline-Score papers, but not for High-Score papers. This indicates that quick and substantive rebuttals can act as a "rescue strategy" for papers at risk, whereas additional responses provide little marginal benefit when scores are already high.

C.4  RQ-I2: ARE MORE EXCHANGES AND GREATER DEPTH SIGNALS OF RESCUE OR RISK?

**Visualization of Interaction Depth and Frequency.**  To further assess the "optimal level" of interaction, Figure 10 presents acceptance rates across binned intervals of response counts and interaction depths.

Table 6: Statistical tests on rebuttal response speed (RQ-I1)

| Group | N | Spearman | Welch t | Logit $\beta$/OR |
|---|---|---|---|---|
| Overall | 20,666 | -0.052 (3.6e-11) | -6.058 (1.4e-09) | -0.103 / 0.902 (3.9e-11) |
| Low-Score-Group | 6,599 | -0.070 (3.6e-07) | -8.511 (5.8e-17) | -0.150 / 0.861 (1.9e-04) |
| Borderline-Score-Group | 8,710 | -0.049 (4.0e-05) | -4.403 (1.1e-05) | -0.096 / 0.909 (7.9e-05) |
| High-Score-Group | 5,357 | 0.004 (7.7e-01) | 1.079 (2.8e-01) | 0.010 / 1.010 (8.6e-01) |

Table 7: Statistical tests on rebuttal response length (RQ-I1)

| Group | N | Spearman | Welch t | Logit $\beta$/OR |
|---|---|---|---|---|
| Overall | 20,666 | -0.059 (4.2e-14) | -11.178 (6.6e-39) | -0.109 / 0.897 (4.8e-12) |
| Low-Score-Group | 6,599 | 0.202 (1.4e-49) | 13.582 (9.9e-39) | 0.988 / 2.686 (1.1e-54) |
| Borderline-Score-Group | 8,710 | 0.181 (3.0e-52) | 9.498 (3.0e-21) | 0.371 / 1.449 (5.6e-52) |
| High-Score-Group | 5,357 | 0.017 2.6e-01) | -0.068 (2.6e-01) | 0.057 / 1.058 (3.3e-01) |

## C.5 RQ-S1: HOW DOES SUBMISSION TIMING (EARLY VS. LAST-MINUTE) AFFECT FINAL ACCEPTANCE?

**Statistical Tests on Submission Timing.** Table 8 statistically confirms these findings. Both Spearman's $\rho$ and Pearson's $r$ show positive correlations between submission timing and acceptance or mean score, while Welch's $t$-test highlights significant differences between early and last-minute groups. In summary, early submission emerges as a strategic factor, linked to higher acceptance, stronger scores, and lower disagreement.

Table 8: Statistical significance tests for submission timing vs. acceptance, mean score, and reviewer disagreement (RQ-S1)

| Group | N | Spearman | Pearson | Welch t (0–1d vs 1–14d) |
|---|---|---|---|---|
| Acceptance | 28,086 | +0.050 (1.23e-15) | +0.058 (4.3e-20) | -8.58 (9.89e-18) |
| Avg score | 28,086 | +0.052 (1.69e-16) | +0.077 (8.6e-35) | -14.05 (1.20e-44) |
| Gini Index | 28,086 | +0.002 (0.697) | -0.037 (5.9e-09) | +7.49 (6.91e-14) |

## C.6 RQ-S3: DOES PROVIDING REPRODUCIBILITY INFORMATION AFFECT FINAL DECISIONS?

**Statistical Tests on Reproducibility Resources.** Table 9 reports statistical tests. Spearman's $\rho = 0.116$ ($p < 1 \times 10^{-80}$), Welch's $t = 19.762$ ($p < 1 \times 10^{-80}$), and logistic regression showed an odds ratio (OR) of 1.626. This demonstrates that RS provision is not merely correlated with, but strongly predictive of, higher acceptance probability. Thus, reproducibility support functions as a decisive trust signal for reviewers.

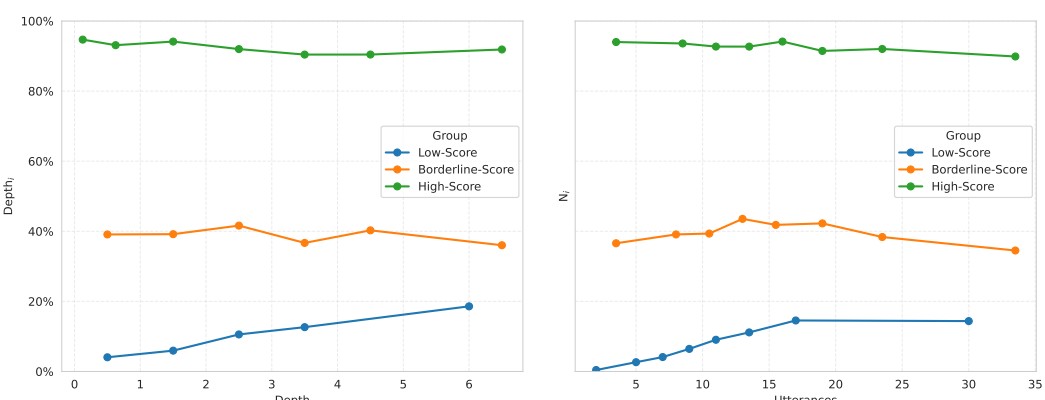

Figure 10: Acceptance rates by depth and frequency of author–reviewer interactions (RQ-I2)

Table 9: Statistical tests for reproducibility resources vs. acceptance (RQ-S3)

| Group | N | Spearman | Welch t-test | Logit $\beta$/OR |
|---|---|---|---|---|
| Overall | 28,086 | 0.116 (2.81e-85) | 19.762 (2.56e-86) | 0.486 / 1.626 (3.19e-84) |

## C.7 RQ-P1: How do visual elements (tables, figures, equations) relate to novelty and contribution?

**Correlation Tests on Visual Elements.** Table 10 reports correlation tests between each visual element (tables, figures, equations) and novelty or contribution scores. The results can be summarized as follows:

- Tables. Table count shows a negative correlation with technical novelty but a significant positive correlation with both empirical novelty and contribution scores. This suggests that while more tables help systematize comparisons and results, they may not signal newness of ideas. Instead, they strongly support empirical novelty and highlight research contribution.
- Figures. Figures are negatively correlated with technical novelty but positively correlated with empirical novelty. That is, papers with many figures may appear less fresh in terms of technical ideas but more convincing in terms of empirical novelty. No clear relationship with contribution scores was observed.
- Equations. Equations exhibit little correlation with novelty scores but a significant positive correlation with contribution scores, indicating that formal mathematical formulation can serve as a signal of research contribution.

## C.8 RQ-P2: How do English fluency and clarity relate to acceptance rates?

**Linguistic Features in the Pre-LLM Era.** In the analysis, clear differences were observed before and after the widespread use of LLMs. Table 11 reports detailed correlations between linguistic metrics and acceptance rates in the Pre-LLM era (2017–2023).

**Pre-LMM (2017–2023).** The analysis shows that in the Table 11, papers that were easier to read were somewhat more likely to be accepted. In addition, a tendency to write slightly longer sentences served, albeit weakly, as a positive signal. More specifically:

- Vocabulary diversity: Higher values were significantly associated with lower acceptance rates. Excessive use of varied vocabulary may have been perceived as reducing the consistency of a paper.
- Average sentence length: Longer sentences were mildly correlated with higher acceptance rates, though the effect size was limited.
- Average words per sentence: This metric had almost no effect on acceptance.
- Flesch Reading Ease: Higher values (i.e., easier text) showed a weak positive correlation with acceptance, though neither the correlation coefficient nor regression significance was large.

Table 10: Correlation tests between visual elements (tables, figures, equations) and novelty/contribution scores (RQ-P1)

| Count | Target Score | N | Spearman | Pearson | Mann–Whitney U | Welch t | Logit $\beta$/OR |
|---|---|---|---|---|---|---|---|
| Table | Technical Novelty | 5,448 | -0.035 (8.87e-03) | -0.034 (1.32e-02) | 3553770.500 (5.90e-02) | -1.768 (7.72e-02) | -0.016 / 0.984 (7.71e-02) |
| | Empirical Novelty | 5,448 | 0.104 (1.32e-14) | 0.123 (6.83e-20) | 4007910.000 (5.55e-08) | 5.281 (1.33e-07) | 0.048 / 1.049 (1.56e-07) |
| | Contribution | 7,192 | 0.093 (3.73e-15) | 0.103 (1.86e-18) | 6978843.000 (8.49e-10) | 6.520 (7.53e-11) | 0.050 / 1.052 (9.98e-11) |
| Figure | Technical Novelty | 4,661 | -0.063 (1.82e-05) | -0.052 (3.40e-04) | 2532290.000 (5.29e-04) | -2.812 (4.94e-03) | -0.028 / 0.972 (4.97e-03) |
| | Empirical Novelty | 4,661 | 0.076 (1.84e-07) | 0.093 (2.24e-10) | 2893593.000 (3.90e-05) | 3.891 (1.01e-04) | 0.039 / 1.040 (1.07e-04) |
| | Contribution | 5,653 | -0.013 (3.40e-01) | -0.009 (5.08e-01) | 3992144.000 (9.79e-01) | 0.291 (7.71e-01) | 0.003 / 1.003 (7.71e-01) |
| Equation | Technical Novelty | 6,387 | -0.007 (5.58e-01) | -0.011 (3.88e-01) | 5042897.000 (9.03e-01) | -0.187 (8.52e-01) | -0.006 / 0.994 (8.51e-01) |
| | Empirical Novelty | 6,387 | 0.020 (1.02e-01) | 0.018 (1.47e-01) | 5097371.000 (1.87e-01) | 1.115 (2.65e-01) | 0.036 / 1.037 (2.69e-01) |
| | Contribution | 8,586 | 0.054 (6.76e-07) | 0.038 (5.03e-04) | 9371215.000 (5.72e-05) | 3.263 (1.11e-03) | 0.088 / 1.092 (1.42e-03) |

Table 11: Correlation between linguistic metrics and acceptance rates in the Pre-LLM era (2017–2023)

| Metric | N | Spearman | Pearson | Mann–Whitney U | Welch t | Logit $\beta$/OR |
|---|---|---|---|---|---|---|
| Vocabulary Diversity | 14,027 | -0.061 (3.12e-13) | -0.060 (1.71e-12) | 21342565.500 (3.27e-13) | -7.032 (2.16e-12) | -1.226 / 0.293 (1.98e-12) |
| Average Sentence Length | 14,027 | 0.046 (3.60e-08) | 0.019 (2.45e-02) | 24310424.500 (3.66e-08) | 2.524 (1.16e-02) | 0.010 / 1.010 (3.76e-02) |
| Average Words per Sentence | 14,008 | -0.005 (5.33e-01) | -0.012 (1.56e-01) | 22833124.000 (5.33e-01) | -1.465 (1.43e-01) | -0.010 / 0.990 (1.56e-01) |
| Flesch Reading Ease | 14,008 | 0.020 (2.03e-02) | 0.005 (5.77e-01) | 23514833.000 (2.03e-02) | 0.547 (5.84e-01) | 0.000 / 1.000 (5.77e-01) |
| Flesch–Kincaid Grade | 14,008 | -0.021 (1.51e-02) | -0.005 (5.71e-01) | 22414881.000 (1.51e-02) | -0.555 (5.79e-01) | -0.001 / 0.999 (5.71e-01) |
| Gunning Fog Index | 14,008 | -0.034 (5.62e-05) | -0.030 (3.58e-04) | 22044638.000 (5.64e-05) | -3.573 (3.54e-04) | -0.027 / 0.973 (3.63e-04) |

- Flesch–Kincaid Grade: Higher values (i.e., more difficult texts) showed a weak negative relationship with acceptance.
- Gunning Fog Index: Higher values (i.e., more difficult writing) were clearly associated with lower acceptance rates.

**Post-LLM (2024–2025).** In contrast, in Table 12, these patterns were reversed. Papers written in a professional and somewhat difficult tone were favored, but at the same time, short and concise sentences were more highly rated. The detailed results are as follows:

- Vocabulary diversity:Higher values were associated with lower acceptance rates. This relationship was statistically significant, though the effect size was small.
- Average sentence length and words per sentence: Longer or wordier sentences were strongly linked to lower acceptance rates. Short and simple sentences were more effective.
- Flesch Reading Ease: Higher values (easier text) were actually associated with lower acceptance rates.
- Flesch–Kincaid Grade and Gunning Fog Index: Higher values (harder texts) correlated with higher acceptance rates.

C.9 RQ-P3: HOW DOES REFERENCE RECENCY INFLUENCE NOVELTY AND REVIEWER EVALUATION?

**Visualization of Reference Recency and Novelty.** Figure 11 visualizes LOWESS trends between reference metrics and novelty scores. The patterns show that older references (higher average age, higher old ratio) correlate with lower empirical novelty, while a higher proportion of recent refer-

Table 12: Correlation between linguistic metrics and acceptance rates in the Post-LLM era (2024–2025)

| Metric | N | Spearman | Pearson | Mann–Whitney U | Welch t | Logit $\beta$/OR |
|---|---|---|---|---|---|---|
| Vocabulary Diversity | 14,331 | -0.023 (5.44e-03) | -0.023 (6.75e-03) | 13138524.500 (5.45e-03) | -2.669 (7.64e-03) | -0.608 / 0.545 (6.77e-03) |
| Average Sentence Length | 14,331 | -0.130 (1.20e-54) | -0.107 (9.91e-38) | 10841862.000 (3.31e-54) | -14.400 (9.92e-46) | -0.105 / 0.900 (8.94e-40) |
| Average Words per Sentence | 14,319 | -0.166 (1.22e-88) | -0.156 (9.35e-79) | 10047479.500 (1.87e-87) | -21.491 (1.65e-96) | -0.184 / 0.832 (3.50e-79) |
| Flesch Reading Ease | 14,319 | -0.092 (1.48e-28) | -0.098 (1.22e-31) | 11627226.000 (1.92e-28) | -10.750 (1.82e-26) | -0.004 / 0.996 (3.49e-30) |
| Flesch–Kincaid Grade | 14,319 | 0.102 (1.36e-34) | 0.105 (4.49e-36) | 15827913.000 (2.01e-34) | 11.617 (1.54e-30) | 0.031 / 1.032 (1.08e-33) |
| Gunning Fog Index | 14,319 | 0.158 (1.38e-80) | 0.131 (7.77e-56) | 17028248.000 (1.31e-79) | 17.415 (3.39e-65) | 0.147 / 1.158 (8.64e-57) |

ences correlates with higher empirical novelty. The correlation with technical novelty is statistically significant but very small in magnitude, thus practically negligible.

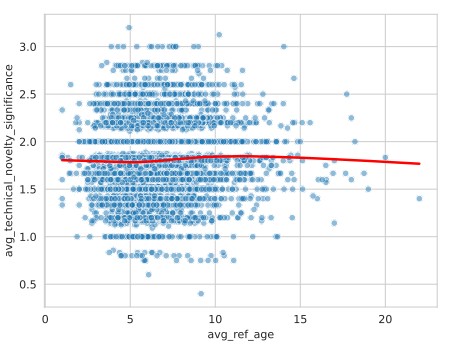

(a) Average Reference Age vs. Technical Novelty

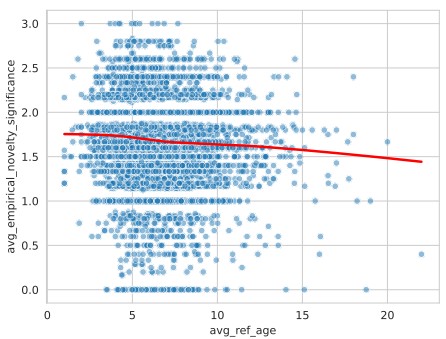

(b) Average Reference Age vs. Empirical Novelty

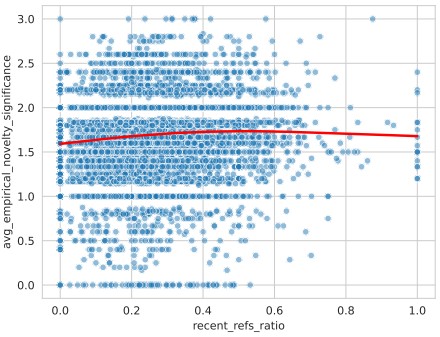

(c) Recent Reference Ratio vs. Empirical Novelty

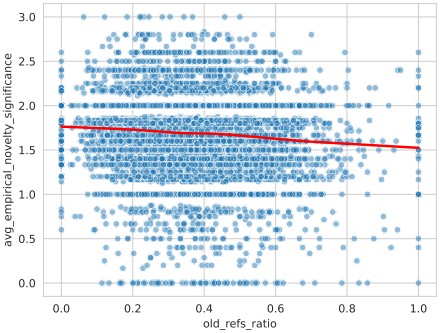

(d) Old Reference Ratio vs. Empirical Novelty

Figure 11: LOWESS trends illustrating how reference metrics (average age, recent ratio, old ratio) relate to novelty scores (RQ-P3)

**Statistical Validation of Reference Recency.** Statistical validation in Table 13 provides further support for these observations and highlights how temporal patterns in citations are systematically linked to reviewer evaluations:

- **Average reference age:** Papers citing predominantly older references show a strong decrease in empirical novelty, indicating that reliance on outdated sources signals limited engagement with recent advances. By contrast, the effect on technical novelty is negligible, suggesting that the age of references matters less for technical soundness than for the perception of novelty.

Table 13: Correlation tests between reference recency metrics and novelty scores (RQ-P3)

| Reference Metric | Target | N | Spearman | Pearson | Mann–Whitney U | Welch t | Logit $\beta$/OR |
|---|---|---|---|---|---|---|---|
| Average Reference Age | Technical Novelty | 6,374 | 0.042 (7.02e-04) | 0.041 (9.28e-04) | 5206185.500 (1.40e-02) | 2.036 (4.17e-02) | 0.024 / 1.025 (4.19e-02) |
| | Empirical Novelty | 6,374 | -0.090 (6.22e-13) | -0.096 (1.60e-14) | 4564970.500 (8.62e-11) | -6.153 (8.09e-10) | -0.074 / 0.929 (8.42e-10) |
| Recent Reference Ratio | Technical Novelty | 6,374 | -0.009 (4.70e-01) | -0.007 (5.57e-01) | 4981259.500 (5.35e-01) | -0.479 (6.32e-01) | -0.082 / 0.922 (6.32e-01) |
| | Empirical Novelty | 6,374 | 0.085 (9.25e-12) | 0.074 (3.43e-09) | 5506066.000 (1.87e-10) | 5.524 (3.46e-08) | 0.952 / 2.590 (3.75e-08) |
| Old Reference Ratio | Technical Novelty | 6,374 | 0.020 (1.19e-01) | 0.015 (2.33e-01) | 5112707.000 (2.39e-01) | 0.824 (4.10e-01) | 0.130 / 1.139 (4.10e-01) |
| | Empirical Novelty | 6,374 | -0.086 (6.64e-12) | -0.093 (1.23e-13) | 4573003.000 (1.77e-10) | -6.467 (1.08e-10) | -1.025 / 0.359 (1.13e-10) |

- **Recent references ratio:** A higher share of recent references significantly boosts empirical novelty. This pattern implies that reviewers regard engagement with the latest literature as evidence of originality and contribution to the state of the art.
- **Old references ratio:** Conversely, a higher proportion of old references significantly reduces empirical novelty. Such reliance on older work may create the impression of incremental rather than innovative research.

Together, these results validate that reference recency is not a peripheral stylistic choice but a substantive signal influencing how novelty is assessed by reviewers.

