# OpenReview forum: "What Drives Paper Acceptance? A Process-Centric Analysis of Modern Peer Review"
_ICLR.cc/2026/Conference — Submitted to ICLR 2026_

### Official Review · Reviewer_pqCu · 2025-10-26

**Soundness:** 3
**Presentation:** 3
**Contribution:** 4
**Rating:** 8
**Confidence:** 3

**Summary:**

This paper performs an in-depth analysis of the decision criteria that lead to a paper being accepted at ICLR conferences. The paper collects an extensive ICLR peer review dataset spanning from 2017 to 2025, with additional sources from ArXiv. The paper first analyzes how the level of disagreement between reviewers affects a paper's acceptance based on three score categories (low, borderline, high). The authors find that disagreement within the low-score group may help acceptance, and that the sentiment of the review itself plays an important role. While these results may have been shared anecdotally through social media platforms, presenting quantitative results in this manner is the first of its kind (to the best of the reviewer's knowledge). The paper then examines the interaction dynamics between reviewers and authors. Interestingly, papers in the low-score group have a higher chance of acceptance when there is more active engagement, whereas papers in the high-score group experience a decreased chance of acceptance with increased engagement. The paper also analyzes whether posting to ArXiv influences acceptance rates, reporting that ArXiv posting has a high correlation with acceptance, as it may indicate that the paper is more mature (though I believe there may be other confounding factors, such as authors coming from large tech companies or well-known research groups). Overall, the paper is of high quality, and the results reported are insightful.

**Strengths:**

- Overall, the reviewer was genuinely impressed by the depth of analysis and the effort invested by the authors. Great work.
-  The selective analysis focusing on areas where it is most necessary (especially on borderline groups) is interesting, and the detailed findings made the paper easy to understand.
- It was fascinating to see a large amount of anecdotal evidence verified with quantitative data. This will be very helpful to researchers.
-  The analysis of whether papers submitted to ArXiv and their reproducibility was notable.
-  The paper has several practical implications. The OpenReview team should consider providing statistics on the variance of reviewer scores and confidence levels for meta-reviewers based on the discussion in Section 7, as an easy visual cue to enhance decision-making.

**Weaknesses:**

-	While the results regarding ArXiv posting were interesting, I wonder whether the findings may be biased by papers that are resubmissions of work rejected from other venues prior to ICLR. Additionally, authors from established research groups or companies are more likely to submit their work to ArXiv compared to researchers from less well-known groups. These limitations should be discussed.

**Questions:**

**Suggestions.**
-  Additional analysis of papers that have been resubmitted to ICLR conferences (e.g., the same paper being submitted to ICLR 2023, rejected, then resubmitted to ICLR 2024) would be an interesting extension. This could potentially be identified through author list overlap and paper content similarity.
-  Can the authors provide a brief explanation of the readability indices used in Section 6.2?
-  While this is clearly not a weakness of the paper itself, as a reviewer, I think it is worth mentioning my concerns regarding how this paper might be used by future researchers. This paper provides an in-depth, data-driven analysis of how authors can optimize their work during the review process to increase their chances of acceptance. However, I am concerned that future researchers might interpret these findings as a prescriptive checklist for acceptance rather than as descriptive insights about the review process. There is a risk that researchers may overly focus on optimizing for acceptance metrics at the expense of pursuing genuinely novel ideas or meaningfully engaging with reviewers to enhance the quality of their work. Research should ultimately be driven by intellectual curiosity and the pursuit of knowledge, not solely by strategic considerations for publication. I highly recommend that the authors include an impact statement addressing these concerns.

**Questions.**
- I believe there is a typo in Line 87: "f Peer Review"
-  How is the submission timing measured? Authors often re-edit their submissions until the last minute. Is the first submission timing used, or the last? An explanation of how this timing is measured would benefit readers.

---

> ### Author Response · Authors · 2025-11-24
> **Rebuttal to Official Review of Submission23540 by Reviewer pqCu**
>
> We sincerely thank the reviewer for the constructive feedback. We've carefully reviewed each of your points, and we're happy to provide the following responses. We've tried to fill in any gaps in our explanation, and to communicate your concerns as clearly as possible.
>
> **Response to Weaknesses**
>
> >Regarding the arXiv-related analysis and potential confounders, our study was conducted using only data directly available from the OpenReview API. As a result, we were not able to directly identify whether a paper had been previously rejected at another venue and then resubmitted to ICLR, and this aspect was therefore not sufficiently discussed. We had also considered incorporating information about whether authors are affiliated with companies or industry labs, but in practice we found that inferring “industrial vs. academic” affiliation reliably from OpenReview author IDs alone was difficult, and we decided that including such a noisy signal in the main analysis would be misleading. We see these points as important limitations of the current work and plan to explore them more thoroughly in future analyses with additional data and matching procedures.
>
> **Response to Suggestions-1**
>
> >The resubmission-related idea you suggested—tracking likely resubmissions across years (e.g., via overlapping author lists and similar titles/content) and re-examining the arXiv–acceptance relationship in that subset—is, in our view, a very natural and interesting extension of our framework. While it was beyond the scope of the present submission, we would like to pursue this direction in follow-up analyses and experiments.
>
> **Response to Suggestions-2**
>
> > For the readability indices in Section 6.2, the measures we use are as follows:
> >
> > - **Flesch Reading Ease**
> >   Quantifies how easy a text is to read, based primarily on average sentence length and syllables per word.
> >   *(Higher scores indicate easier text.)*
> >
> > - **Flesch–Kincaid Grade**
> >   Uses similar statistics as above but converts them into an approximate U.S. school grade level required to understand the text.
> >
> > - **Gunning Fog Index**
> >   Estimates how many years of formal education a reader would typically need to understand the text on first reading, using sentence length and the proportion of complex words.
>
> In the revised version, we will include a brief explanation along these lines and clarify that these indices are used solely as simple proxies for relative lexical/sentence complexity, rather than as absolute measures of *“good writing.”*
>
> **Response to Suggestions-3**
>
> > We also share the concern that our findings could be misinterpreted as a prescriptive checklist for getting a paper accepted. We will adjust the tone throughout the discussion and impact statement to avoid that impression, reducing any overly directive “should”-style language. Our goal is to present a descriptive, process-centric analysis of observable patterns in the review workflow, not to provide a recipe for acceptance, and we will make this intention more explicit.
>
> **Response to Questions-1**
>
> > Thank you. The “f Peer Review” on Line 87 you pointed out is indeed a typo.
> We apologize for not catching this in the submitted version and will correct it to the accurate expression in the camera-ready version.
> Thank you once again for your careful review.
>
> **Response to Questions-2**
>
> > Finally, regarding submission timing: we measure timing using the first submission time of each paper, i.e., the timestamp when the note was initially created on OpenReview (not the last pre-deadline revision). We agree that this definition was not clearly stated and will add an explicit explanation to the data description and the relevant results sections.

---

### Official Review · Reviewer_hUNp · 2025-10-29

**Soundness:** 1
**Presentation:** 1
**Contribution:** 1
**Rating:** 0
**Confidence:** 4

**Summary:**

The paper presents itself as an analysis of ICLR (2017-2025) paper submission data at a “process level.”   The analysis takes into account “reviews, meta-reviews, rebuttal and interaction threads, reviewer scores and confidence, submission and deadline timestamps, arXiv postings, and code/data disclosures” as well content extracted from pdfs of paper submissions.  The findings of this analysis include:

-	For papers with higher and mid-level (“borderline”) average scores, more disagreement among reviewers tends to mean rejection, but for papers with lower average scores, the reverse.  The paper makes no attempt to explain the mechanism.

-	Connection of self-reported reviewer confidence to acceptance is judged to be null in borderline cases.  Sentiment analysis is used to argue that, for borderline papers, negative comments are damning while positive ones are only slightly helpful relative to the base acceptance rate.

-	Lower latency of authors to provide rebuttals and greater length of reviewer messages are associated with higher acceptance chances for low and borderline papers (null effect for high scoring papers).

-	Deeper message threads and more messages from authors in rebuttal period correlate with higher acceptance rates for low scoring papers, perhaps slightly negative or neutral for borderline/high papers.

-	Earlier submissions tend to show higher scores, higher chance of acceptance, and lower disagreement.

-	Posting the paper on arXiv, especially after submission time, correlates with higher chance of acceptance.

-	Including “reproducibility-supporting” materials correlates with higher chance of acceptance.

-	More tables and equations correlate with higher empirical contribution scores.

-	Post ChatGPT release (2023), the pattern of correlations between simple stylistic measures and acceptance shifts somewhat; the paper describes this as a shift from a preference from “easier-to-read papers with longer sentences to ones that use shorter sentences and a more “professional” tone.

-	Papers whose citations skew to more recent tend to receive higher novelty scores.

**Strengths:**

-	The paper promises to release the dataset, which would be valuable for the community, especially if paired with all the code used to process the data (to reproduce and improve upon the calculations done in this study).

**Weaknesses:**

-	The paper makes extensive use of causal language, but it is not a causal analysis.  All of the conclusions are based on correlations of one kind or another, with no attempt to explore mechanisms that explain the correlations and lack thereof.  There are many assumptions at work here, and considerable missing information about the peer review process, which are not called out in the paper.  E.g., we only see a reviewer’s final assessment, we don’t know how carefully they read the paper, how much of an expert they are, or how experienced they are with reviewing in this community.
-	The paper doesn’t engage with prior literature relevant to the questions being asked; after the introduction, nothing is cited (there is a related work section, but it’s tucked in the appendix).  Notably, other datasets published by others in the community are not discussed, e.g., https://api.semanticscholar.org/CorpusId:13746581 … more examples of missing connections to literature below.
-	Signficance tests are given in the appendix only; there are many of them and I don’t see a discussion of correction for multiple hypothesis tests across the study.
-	Consistently throughout the paper, papers are grouped by their average scores (low, borderline, high, with the groups defined by the ICLR review ratings).  There is no robustness check for whether the findings hold up if we shift these borders.
-	Section 3 is basically about modeling meta-reviewer/area chair + program committee decisionmaking.  But it makes no reference to these actors except at the end (line 236), mentioning  “the need for meta-reviewers to balance quantitative signals (score variance) with qualitative cues (review sentiment) when mediating reviewer disagreements.”  This remark confuses the underlying construct (decisionmaking) with the operationalized quantities (variance of scores and model-derived sentiment classification).
-	The sentiment analysis approach is not described other than pointing to a Huggingface model.  The authors do not give a justification for this choice of model and do not validate it on paper reviews (most sentiment analysis models are trained on a completely different domain).  Most problematically, sentiment is carried out at the whole-review level.  But reviews tend to include many points, both positive and negative, and what’s really at work is the process by which an expert considers each point and ultimately weighs it in a final decision for the paper.
-	Section 4:  there is no discussion of any past work on analyzing logged conversations; authors use simplistic, easy-to-measure quantities like average message length and depth of conversation thread, without justifying them.  Example of recent work that is more careful:  https://api.semanticscholar.org/CorpusId:235186920
-	Paper lacks any discussion of temporal shift in the process, apart from a change in style/acceptance patterns of correlation that is uncritically attributed to the release of ChatGPT in 2023.  This was a period of tremendous growth for the field, and likely considerable cultural change, including in the review process.
-	The authors do give a mechanism for the research question about intensity of interaction and chance of acceptance.  But the mechanism – that too much engagement tires out reviewers and leads to negative outcomes – doesn’t actually align with the finding that more engagement correlates with better acceptance chances for low-scoring papers.  It seems that the patience of the decisionmakers (not the reviewers) is actually more important here, as well as the qualitative content of those back-and-forths.
-	Recommendations from section 4 seem consistent with almost anything that could have been found in the data:  “Authors must tailor strategies to their score group, while meta-reviewers should weigh not only quantitative indicators but also their contextual meaning.”  I’m trying to imagine a finding that would be inconsistent with these recommendations.
-	Timing of submissions:  it’s not discussed is whether it’s the final submission time of the paper, or the first submission time.
-	An undiscussed factor in submission timing is whether the paper has gone through peer review before; resubmissions are often early, whether or not the authors have taken into account previous reviews.  This might be technically challenging data to collect, but it should nonetheless be discussed.
-	Section 5.2, about how arXiv posting affects acceptance, doesn’t cite a single paper on this topic, but there are many that engage much more deeply with the question and the underlying causal factors than this one does, for example: https://api.semanticscholar.org/CorpusId:220280205 https://api.semanticscholar.org/CorpusId:44222876 https://api.semanticscholar.org/CorpusId:232170561 https://api.semanticscholar.org/CorpusId:247839500 https://api.semanticscholar.org/CorpusId:13746581
-	Section 6 makes use of a package called pdfplumber for extracting visual elements from paper pdfs.  The quality of this tool is not assessed, and there is no discussion of its technical approach.  The mention of “parsing errors” without quantification or discussion does not inspire confidence.
-	Fluency and readability are operationalized using vocabulary diversity (unique-to-total word ratio),  average sentence length, average words per sentence, and readability indices from the 1940s through 1970s), which are cited without comment.  There is extensive literature on style in the past 50 years; a recent survey (focusing on a different application of detecting AI generated text, but relevant nonetheless) is https://api.semanticscholar.org/CorpusId:268230672   From these metrics alone, the authors claim conclusions about “easier-to-read” and “more professional” writing.
-	Changes in correlation patterns between style variables and acceptance are attributed to ChatGPT and language model use without considering alternatives.  Perhaps this trend was evident all along due to rising time pressure on reviewers in a growing community (hinted at line 471, “time constraints”).
-	Despite the marketing as a “process level” analysis, the paper consists entirely of simple correlations between easily measured variables, pairwise.
-	Line 484 – “a foundation for fairer, reproducible practices” – there is no evidence to support that the recommendations in this paper have anything to do with fairness or reproducibility.

**Questions:**

-	Value ranges for ratings and confidence are not given.  Were they the same for all iterations of ICLR?  In general there is no discussion of any changes in ICLR review forms or review instructions, but surely they did change over the time under study.  Does it really make sense to use means and variances for ordinal data like this?

-	Line 258 “Differences between groups are statistically significant.” – what are the groups?  Why measure length of reviewer messages and latency of author messages, rather than latency and length of both?

-	Line 374:  “The novelty, contribution, and experimental results of a paper are the primary factors reviewers consider when evaluating its quality.”  Where is the evidence for this?  If this is a statement about the ICLR reviewing forms, that should be made plain.  As written it’s a statement about how reviewers behave and requires evidence.

-	The paragraph at line 386 is extremely confusing, discussing technical and empirical contributions and novelty altogether without teasing apart these different concepts.  Are technical and empirical novelty the same or different?

---

> ### Author Response · Authors · 2025-11-24
> **Rebuttal to Official Review of Submission23540 by Reviewer hUNp**
>
> We sincerely thank the reviewer for the constructive feedback.
> To respond clearly, we group the issues by theme and provide consolidated answers.
>
> **Regarding *“this is merely simple correlation analysis”***
>
> > We acknowledge that our results are based on correlation analysis, and some phrasing may have unintentionally implied causality. True causal analysis would require controlled experiments for each RQ, which we did not conduct due to scope and space constraints. Thus, we agree that causal effects cannot be claimed.
> >
> > Still, several patterns show statistically significant signals, and our goal was to present them in an accessible way. We agree that some sentences were too definitive and will revise them to avoid implying causal conclusions.
>
> **Regarding requests to analyze data not provided by the OpenReview API**
> > Our study uses only publicly available OpenReview API data, focusing on questions researchers commonly have when submitting or reviewing papers. We conducted analyses strictly within what the API provides.
> >
> > As noted by the reviewer, reviewer expertise, background, experience, meta-reviewer behavior, and resubmission status are not observable. We intentionally avoided inferring such unobservable factors to prevent bias. We see analyses incorporating these richer signals as valuable directions for future work.
>
> **Regarding lack of related work**
> > We acknowledge that the coverage of related work is insufficient. Due to space constraints, we selected only a few representative works. The prior studies the reviewer points out are, in many cases, intense and focused analyses on each specific subproblem or RQ in their own scope. In contrast, our goal was to cover these aspects across the full ICLR corpus and perform a broader analysis to understand overall trends. From this perspective, we believe that the quantitative analysis in our paper is not weak in volume, but rather complementary in breadth.
> >
> >That said, we agree that the connections to this literature should be strengthened, and we will revise the manuscript to incorporate and discuss the suggested works more thoroughly.
>
> **Other specific points to be clarified or improved**
> - *“Timing of submissions: it’s not discussed whether it’s the final submission time of the paper, or the first submission time.”*
>
>     → We use the **first submission time**. We will clarify this in the revised version.
>
> - Lack of discussion of pdfplumber’s limitations.
>
>     → Due to time and space constraints, our explanation of this part was indeed insufficient. We will expand the description of how pdfplumber is used and where its limitations lie.
>
> - *“Line 484 – ‘a foundation for fairer, reproducible practices’ – there is no evidence to support that the recommendations in this paper have anything to do with fairness or reproducibility.”*
>
>     → This wording was motivated by the fact that we intended to release our code and that most of the data we use are accessible via the OpenReview API, which together can support reproducible analysis on the review process. Once our code release is finalized and better documented, we believe this aspect can be strengthened; we will adjust and refine this statement in the meantime.
>
> - Value ranges for ratings and confidence.
>
>     → Rating is on a 1–10 scale and confidence on a 1–5 scale, and this has been consistent from 2017 to 2025 [1][2]. Our description was clearly not explicit enough; we will make sure to state these ranges clearly.
>
> - Robustness of low / borderline / high score groups.
>
>     → In the data, the “recommendation” field explicitly defines 5 as “marginally below” and 6 as “marginally above” the acceptance threshold, so the data itself marks these as borderline scores. We therefore used 5 and 6 to define the borderline group, which we believe is a principled choice aligned with ICLR’s guidelines. While robustness checks with alternative thresholds are useful, our primary grouping follows the official semantics in the data.
>
> Overall, our paper does not aim to probe each detailed RQ in an intense or narrow way, but rather to take the OpenReview API data and systematically analyze a range of questions that many researchers are curious about, confirming which patterns are statistically significant and then distilling messages that readers can easily take away.
>
> We understand that this may differ from the depth the reviewer ideally expects for each individual RQ, but we believe our work serves as a useful starting point: it provides a broad, quantitative map of the process-level landscape and can act as a foundation on which more intensive, narrowly focused follow-up studies—such as those cited by the reviewer—can be built.
>
> > [1] Wang, G., Peng, Q., Zhang, Y., & Zhang, M. (2023). What have we learned from OpenReview?. World Wide Web, 26(2), 683–708.
> >
> > [2] https://papercopilot.com/statistics/iclr-statistics/iclr-2025-statistics/

---

### Official Review · Reviewer_mWnr · 2025-10-30

**Soundness:** 3
**Presentation:** 3
**Contribution:** 2
**Rating:** 4
**Confidence:** 5

**Summary:**

This paper presents a large-scale, process-level analysis of over 28,000 ICLR submissions from 2017 to 2025, focusing on how factors beyond core paper content—such as reviewer disagreement, author-reviewer interactions, external metadata (e.g., arXiv, reproducibility resources), and stylistic elements—influence acceptance decisions. By mining OpenReview logs, arXiv metadata, and PDF-derived features, the authors study signals across four dimensions: (1) reviewer disagreement and confidence, (2) rebuttal interaction patterns, (3) submission strategies and openness, and (4) writing quality, visual layout, and reference recency.

However, a critical point remains unaddressed regarding the granularity and source of rebuttal interaction data, especially how day-by-day interaction timelines were captured—OpenReview and arXiv do not natively expose such temporal resolution.

**Strengths:**

* Covers the full ICLR review corpus over 9 years, with multi-faceted features: scores, discussions, rebuttals, writing style, submission timing, arXiv metadata, and layout.
* Offers actionable recommendations (e.g., early submission, concise rebuttals, clear layout) backed by robust statistical tests (Pearson, Mann–Whitney, t-tests).
* Well-structured methodology: Each research question is backed with statistical analysis—Spearman, Pearson, Mann–Whitney U, Welch’s t-test, logistic regression—adding robustness.

**Weaknesses:**

* Critical data sourcing gap: The paper does not explain how rebuttal dynamics (speed, daily interaction, back-and-forth timing) were collected, especially per-day changes. OpenReview API does not expose the timestamped edit logs for score in a granular way. This is a key missing link in the entire rebuttal dynamics analysis.
* Causal claims from correlational data: While the study acknowledges this, several conclusions (e.g., "response depth hurts High-Score papers") are phrased strongly without causal justification.
* Limited generalizability: The dataset only covers ICLR. The review process, rebuttal length, and acceptance criteria can vary widely across CVPR, NeurIPS, ACL, etc.

I'd love to raise my score if the data sourcing is clearly addressed in this work.

**Questions:**

* Where exactly were the timestamps for rebuttal interactions (e.g., response speed, dialogue depth, score changes) sourced? Were they inferred from OpenReview API created and modified fields? If so, was this normalized across years and reviewers with inconsistent usage?
* Were there any preprocessing steps to exclude reviewers who didn’t engage at all?

---

> ### Author Response · Authors · 2025-11-24
> **Rebuttal to Official Review of Submission23540 by Reviewer mWnr**
>
> We sincerely thank the reviewer for the constructive feedback. Below, we provide point-by-point responses to each weakness and question, followed by a dedicated Data Sourcing section as requested.
>
> **Response to Weaknesses-1 *"Critical data sourcing gap"***
> >We agree that the submission did not sufficiently clarify how rebuttal dynamics (response speed, daily activity, thread depth) were computed.
> >Our analysis relies exclusively on note-level timestamps provided by the OpenReview API, which include:
> >
> > - note_id
> > - reply_to (thread linkage)
> > - cdate – user-visible creation timestamp (UNIX ms)
> >
> >Using these fields, we reconstructed the reviewer–author discussion tree and computed response intervals.
> >However, OpenReview does not provide:
> >
> > - edit logs for score changes
> > - text diff logs
> > - deleted message histories
>
> Therefore, our analysis uses only the final, visible notes available through the API.
> We will make this explicit in the camera-ready version.
>
> **Response to Weaknesses-2 *"Causal claims from correlational data"***
> >We acknowledge that our results are based on correlation analysis, and some phrasing may have unintentionally implied causality. True causal analysis would require controlled experiments for each RQ, which we did not conduct due to scope and space constraints. Thus, we agree that causal effects cannot be claimed.
> >
> >Still, several patterns show statistically significant signals, and our goal was to present them in an accessible way. We agree that some sentences were too definitive and will revise them to avoid implying causal conclusions.
>
> **Response to Weaknesses-3 *"Limited generalizability (ICLR-only)"***
> >In our study, we focused on ICLR because it is one of the few major conferences whose review data are publicly accessible through OpenReview over multiple years, enabling a consistent process-level analysis. We acknowledge, however, that results observed in ICLR may not generalize to other venues. We will clarify this limitation in the camera-ready version, and we plan to examine conference-specific patterns in future work as additional multi-year data from other venues become available.
>
> **Response to Questions-1**
>
> >All timestamps were taken from the OpenReview API:
> > - Rebuttal interaction timing → cdate of each review and reply
> > - Modification time (tcdate) recorded but not used
> > - Score changes → not available in the API and therefore not used
> >
> > ***Normalization across years***
> >
> > We manually normalized year-specific schema differences across 2017–2025.
> >
> >These discrepancies—including changes in field names and scoring rubrics—required substantial manual harmonization. All unified schemas are documented in Appendix B.4, and the camera-ready version will additionally include brief mapping examples for clarity. Further explanation is also provided in the Data Sourcing (Ours) section below.
>
> **Response to Questions-2**
> >We excluded reviewer entries that did not represent real participation:
> > - No review score
> > - No review comment
> > - Incomplete review record not tied to a final decision
> > - Withdraw / Desk Reject / Reject-without-review cases
> > Only reviewers who actually participated in the review or rebuttal phase were included.
>
> **Data Sourcing (Ours)**
>
> ***1. Rebuttal Activity Data***
>
> > We extracted all fields provided by the OpenReview API, but used only those consistent across all years for quantitative analyses. Partial fields were excluded, while fields with different names but identical meaning were manually merged into our unified schema. In short, we sourced all usable metadata, adding fields when normalization was possible. Further details are provided in the section below.
> >
> > Not available and not used:
> > - score edit logs
> > - revision histories
> > - deleted messages
>
> ***2. Timestamp Definitions***
>
> > - Submission timing = first submission time (initial cdate)
> > - Rebuttal timing = cdate of review/comment/reply
> > - Score-change timing = not available → not used
>
> ***3. Year-to-Year Schema Normalization***
>
> >ICLR 2017–2025 uses multiple incompatible metadata schemas, requiring manual standardization.
> >
> > Examples of differences:
> >
> > | Year        | Reviewer Detailed Score Fields                                                         |
> > |-------------|----------------------------------------------------------------------------------|
> > | 2022–2023   | Correctness, Technical Novelty & Significance, Empirical Novelty & Significance |
> > | 2024–2025   | Soundness, Presentation, Contribution                                            |
> >
> > Additional differences existed in:
> > - field names
> > - data types
> > - existence of optional metadata (e.g., metareviews)
> > - etc.
> >
> > Normalization steps:
> > 1. Automatic mapping of common fields
> > 2. Manual inspection of all unmatched fields
> > 3. Integration into a unified schema (Appendix B.4)
> >
> The camera-ready version will add explicit examples, mapping tables, and preprocessing scripts.

---

> > ### Comment · Reviewer_mWnr · 2025-11-25
> >
> > My concern is resolved. I believe that when the temporal score changes are also included in the main manuscript will be more supportive. I'll raise my score.

---

### Official Review · Reviewer_QoDh · 2025-10-31

**Soundness:** 1
**Presentation:** 3
**Contribution:** 2
**Rating:** 0
**Confidence:** 4

**Summary:**

The authors of this work perform a study of a large scale data set capturing the peer review process of more than 28.000 papers submitted to ICLR between 2017 and 2025. A focus of the study is to identify determinants of paper acceptance, i.e. the authors address the question whether signals either in the submitted works or the peer-review process can be used to explain the later acceptance or rejection of the paper.

To address this question, the authors consider the following aspects using extensive log data from OpenReview: (i) data on reviewer scores, confidence and sentiment of comments (extracted via a pretrained BERT model), (ii) data on the rebuttal process, (iii) information on submission timing, preprint and code/data availability, and (iv) mansucript characteristics such as linguistic aspects of the text, tables and figure placement.

The key findings of the analysis are:
- Disagreement between reviewer scores influences paper acceptance
- Comments with negative sentiment negatively influence paper acceptance while positive sentiment mildly increases acceptance
- Earlier submissions and submissions with a prior arXiv preprint and reprocibility resources are more likely to be accepted.
- There is a positive correlation between the inclusion of visual elements (tables vs. figures) and the novelty and contribution scores.
- There are (weak) correlations between metrics on textual characteristics and acceptance rates that differ between the pre- and post-LLM era.
- There is a correlation between reference novelty and the (empirical) novelty score as assigned by reviewers.

In summary, the work addresses an interesting problem based on a potentially interesting data set. It contributes to a better quantitative understanding of peer review processes in the ML community but, considering the weaknesses outlined below, it falls short to adequately address the research questions outlined in the paper. Apart from considering a data set on the ICLR peer review process, it is also not clear how the work fits the scope of ICLR.

**Strengths:**

[S1] Quantitatively studying the peer review process of ICLR based on a large data set, this work addresses an important issue - and concerns regarding review quality - that are of interest to researchers in the wider machine learning community.

[S2] The work provides a new data sets collected from OpenReview that could facilitate future research into peer review quality.

[S3] The paper is well-written, well-motivated and formulates a clear set of research questions that are backed by quantitative analyses.

**Weaknesses:**

[W1] The analysis is in my view rather shallow and descriptive. It misses confounding factors and thus does not always address the research questions listed by the authors. See my detailed comments below.

[W2] In terms of methodology, some questions could have benefited from a more rigorous statistical analysis, e.g. performing a linear or logistic regression analysis with control variables and with an assessment of the significance of different factors. Some results are rather descriptive and the significance of the differences, e.g. in percentages or correlation scores, is not always clear. See my detailed comments below.

[W3] Considering the weaknesses in W2, the recommendations distilled in section 7 are not really well-founded and should be considered with caution. Also, it is unclear how the recommendations are backed by the analysis at all, as the authors clearly state that their results are correlational rather than causal.

[W4] Although I leave the final assessment to the area/program chairs, in my view this work does not really fit the scope of ICLR, as it neither addresses machine learning methodology or implications, nor the application of machine learning in a given domain. It's main relation to ICLR is that it analyzes data on the ICLR peer review process. I would argue that a revised (see below) and extended version of this work would rather be a timely and interesting contribution for a journal in the context of "science of science", such as Scientometrics or Infometrics. See also my question about the clarification of relevance for ICLR below.

**Questions:**

[Q1] It would be great if the authors could clarify how their paper relates to the scope of ICLR as summarized in the Call for Papers. If the main contribution relevant for ICLR is the provision of a new data set, it would be helpful to better describe the structure and potential of this data set, and how it could be used to evaluate machine learning tasks in the main manuscript. Also, it would then be important to include more details on the data collection process to be able to assess the contribution of this work.

[Q2] I was not convinced by the interpretation of Figure 2. I agree that low score papers have higher variance but did the authors correct for the number of reviews, i.e. could it be that there is a correlation between the number of reviews for a paper and its acceptance probability? Especially since the variance is calculated for a very small number of scores, this may be important to consider. I would be interested to hear your thoughts on this.

[Q3] The analysis how sentiment influences acceptance is interesting but the results are also somewhat uninformative as the authors did not correct for the reviewer scores. In a nutshell, we would expect that sentiment is strongly correlated with reviewer scores (and thus - hopefully - paper quality). I think it would be more interesting to use paper scores as a confounder and perform a regression analysis, which could shed light on how the sentiment of comments specifically influences paper acceptance or rejection (for papers having the same scores). Do you agree that this would be more interesting?

[Q4] I have a similar question about the analysis of the rebuttal process. Here we would again expect that the length and depth of the responses is correlated with the reviewer scores. In particular, authors that get very low scores in general may be less likely to spend a lot of effort in the rebuttal process, compared to authors of works with generally positive scores, which may try to convince one skeptical reviewer. As such, the analysis in the paper does not really isolate the effect of the rebuttal process, which means it does not really address the causal wording of RQ-I1. Wouldn't it be more meaningful to isolate this effect? Also, do we have any idea whether the differences in the average metrics (depth, etc.) between accepted and rejected papers are significant?

[Q5] I think there is a possible confounder in the analysis for RQ-S1 that is not mentioned in the paper: For last minute submissions, it is more likely that these are first submissions to a conference, while for earlier submissions it is more likely that those are revised papers that have already gone through a peer review process at another conference. This particularly holds for ICLR due to the timing of the ICLR deadline (which is shortly after the NeurIPS notification deadline). I understand that this is difficult to assess, but it should at least be mentioned in the interpretation. The same may hold for a prior arXiv posting (see my additional question in Q7).

[Q6] There have been previous analyses (also of ICLR data from OpenReview) which indicate significant community-differences w.r.t. peer review practices and acceptance rates (see e.g. interactive analysis here [1]). Given the thematic breath of ICLR, I thus believe that the community (e.g. computer vision, NLP, explainability, graph learning) could be an importance factor that the analysis should correct for.

[Q7] The analysis of the effect of arXiv postings is interesting, but it may indicate a different mechanism that is not mentioned. For papers available on arxiv it is easy to identify the authors. This could - depending on the recognition of the authors - have both a positive or negative effect on acceptance. Have you considered this question?

[Q8] I did not fully get the message of Figure 8, which reports correlation scores for a large number of textual metrics. To me, it seems that correlations (at least before LLM) are very weak and it is unclear whether there is a significant effect at all. I understand that there seems to be a larger spread post-LLM but I failed to understand what is the hypothesis for this difference and how it affects the correlation coefficients. Also, a major issue here may be the comparison across different sample sizes, as probably the majority of submission in the corpus are from the last editions of ICLR.

[Q9] Could the authors explain the semantics of the empirical novelty vs. technical novelty scores?

[Q10] I would argue that the fact that section 7 makes behavioral recommendations is in conflict with the authors' statement that the analysis is purely correlational rather than causal (which I agree with). In any case, the authors could get a step closer to making recommendations by incorporating some of the suggestions above.


[1] https://papercopilot.com/statistics/iclr-statistics/iclr-2025-statistics/

---

> ### Author Response · Authors · 2025-11-24
> **Rebuttal to Official Review of Submission23540 by Reviewer QoDh**
>
> We sincerely thank the reviewer for the constructive feedback. We've carefully reviewed each of your points. We've tried to fill in any gaps in our explanation, and to communicate your feedbacks as clearly as possible.
>
> **Response to Weaknesses-1**
> > We agree that our analysis is correlational and observational in nature.
> Some phrasing may have unintentionally sounded causal, and we will revise such expressions.
> True causal inference would require controlled, per-RQ ablations, which were beyond the scope of this work.
> We will explicitly clarify this limitation and ensure that all conclusions remain strictly correlational.
>
> **Response to Weaknesses-2**
> > We agree. During the rebuttal period, we will add:
> >
> > - Reviewer-count–stratified variance analysis for Fig. 2
> > - Additional regression-style comparisons for selected RQs (where feasible within rebuttal time)
> >
> > These additions will help strengthen the statistical rigor without expanding beyond the allowed scope.
>
> **Response to Weaknesses-3**
> > We acknowledge this feedback.
> > Section 7 summarizes descriptive trends, not causal prescriptions.
> We will refine the tone to reflect this explicitly and remove any language that could suggest causal recommendations.
>
> **Response to Weaknesses-4**
>
> > We agree that extending the analysis to NeurIPS, ACL, and CVPR would yield broader insights, and we intend to pursue such venue-specific extensions in follow-up work.
>
> **Response to Questions-1**
> > We clarify our contributions as follows:
> >
> > (1) Construction of a unified, multi-year process-level dataset (2017–2025)
> > - ICLR is the major ML venue with 9 years of fully public OpenReview data. We collected, cleaned, and manually normalized year-to-year schema differences (e.g., changes in reviewer forms, metadata fields, missing values, structural inconsistencies). This unified dataset provides a reusable foundation for future ML research on peer-review dynamics.
> >
> > (2) A data-driven framework for analyzing ML review workflows
> > - We provide an empirical methodology for analyzing review-process signals (disagreement, rebuttals, submission strategy, writing features).
> >
> >Additional sourcing details will be added during the rebuttal,
> and the full preprocessing scripts will be released in the camera-ready version.
>
> **Response to Questions-2**
> > Not yet—but we agree it is necessary.
> We will add reviewer-count–stratified analyses during the rebuttal to confirm whether the variance–acceptance relationship holds after adjustment.
>
> **Response to Questions-3**
> > We agree that full control is difficult.
> As a partial remedy, we compared sentiment within score-stratified groups
> (Low-Score, Borderline-Score, High-Score).
> Meaningful correlations appear consistently within these strata,
> suggesting that sentiment trends are not solely driven by score differences.
> We will clarify this design in the revision.
>
> **Response to Questions-4**
> > We agree that authors with low initial scores may invest less in rebuttals,
> while high-score submissions may focus on convincing a single skeptical reviewer.
> Our analysis does not isolate rebuttal causality, and we will explicitly remove any causal wording in RQ-I1.
>
> **Response to Questions-5**
> > We agree that prior-review history is a plausible confounder.
> OpenReview data does not reveal prior submissions, so we could not test this directly.
> However, this is an important direction, and we plan to explore NeurIPS→ICLR resubmission matching in follow-up work.
>
> **Response to Questions-6**
> > ICLR covers diverse subfields with different acceptance cultures.
> Due to schema variability and missing topic metadata,
> we were not able to reliably extract domain categories.
> We will label this as a limitation and plan for a domain-aware analysis in future work.
>
> **Response to Questions-7**
> > A valid and important point.
> Author identifiability may influence reviewer perception in ways that OpenReview data cannot disentangle.
> As noted in prior responses, examining author-overlap–based resubmission trajectories (across years or venues) is a natural extension.
> We intend to explore this direction in follow-up analyses.
>
> **Response to Questions-8**
> > We will refine the interpretation and add supplementary analyses during rebuttal.
> >
> > - Pre-LLM correlations are indeed weak
> > - Post-LLM variance increases, but this may reflect noise
>
> **Response to Questions-9**
>
> > These two fields come directly from the official ICLR 2022–2023 reviewer form:
> >
> > - Technical Novelty & Significance: Novelty/importance of the methodological contribution.
> > - Empirical Novelty & Significance : Novelty/importance of the experimental evidence (datasets, evaluation, empirical findings).
>
> **Response to Questions-10**
>
> >These were intended only as descriptive observations, not prescriptive guidance.
> We will revise the language to avoid any implication of causality and to ensure the section reads as a descriptive summary only.

---

### Meta-Review · Area_Chair_eVGj · 2026-01-07

**Summary:**

This paper analyses the wealth of data provided by the openreview platform of the review process for ICLR over the years to understand in  a data-driven way what factors contribute to paper acceptance. It is a large scale empirical study that analyses many aspects of the process of review.  The reviews were quite mixed with 8, 4, 0, 0.  The reviewer who gave a 4 signaled in chat that they were convinced by the author response and had raised their score.  Thus, the reviews indeed seemed bi-modal with two accepts and two rejects (albeit one accept may have been borderline).  However, the average score would still be in the negative unless the two 0's were to change significantly change their scores.

The major concerns of the two reviewers seem to relate to the rigor of the statistical analyses and relate to whether the causal claims made by the authors.  One of the reviewers questioned whether the topic of the paper fell within the scope of ICLR.  A major concern of the two low scoring reviewers seem to be whether causal conclusions can be justified from the analysis or rather that the authors have surfaced correlations with many potential confounders.

I believe that bi-modal scores can often be a sign of a particularly interesting paper, which could drive a lot of discussion in the community.  However, the low scores in this case seem indicate to me that the reviewers feel that the paper is not quite ready for publication.  The analysis is not quite there yet to make significant claims about the process of review and what actually drives acceptance.  I worry that presenting correlations with acceptance, which have many unresolved confounders, could increase the noise in the review process (i.e. giving authors something to try to optimize rather than the quality of their research).  At the same time, such data could be helpful to organizers and reviewers to try to minimize the effect of causal or correlated things that are not related to the quality of the work.

The two very low scores indicate to me that the paper could be made stronger, and it seems like a disservice to publish the work when it's not quite ready.  Therefore, the recommendation is to reject the paper.  However I would strongly encourage the authors to continue to develop the work since it seems quite promising and useful (when sufficiently rigorous and causal claims can be made).

**Reviewer Concerns:**

A main argument made by the authors argued the point of whether the paper fell within the scope of ICLR.   I would agree that it does.  I think that any analyses of the ICLR review process over many years of ICLR is certainly of interest to the community, particularly as a position paper.

However, I don't think that the major concern of the two reviewers was scope or fit for ICLR.  Instead, they brought up many questions or issues with the analysis and claims presented in the paper.  A particular concern is whether the observations allow one to make claims of causal relationships to paper acceptance (as the paper title would suggest) or instead are correlated with acceptance.  One of the reviewers is clearly stating that they believe it is the latter.

In the author response the authors addressed point by point the concerns of the two low-scoring reviews.  In many cases, the authors say they will revise the work in the rebuttal stage but it's not clear to me that they have made those changes.

**Reviewer Scores:**

I think the question of scope could be resolved and that would perhaps increase the two 0's by one or two points.  However, I don't believe that the point-by-point responses of the authors would sway these two reviewers significantly, particularly because it's not clear that the authors have made changes to the paper to address these points.

---

### Decision · Program_Chairs · 2026-01-26

Reject